# Identification of intracellular cavin target proteins reveals cavin-PP1alpha interactions regulate apoptosis

Kerrie-Ann McMahon[1,7], Yeping Wu [1,7], Yann Gambin[1,4], Emma Sierecki[1,4], Vikas A. Tillu [1], Thomas Hall [1], Nick Martel[1], Satomi Okano[1], Shayli Varasteh Moradi[1,5], Jayde E. Ruelcke[2], Charles Ferguson[1], Alpha S. Yap[1], Kirill Alexandrov[1,6], Michelle M. Hill [2] & Robert G. Parton [1,3]

Caveolae are specialized domains of the plasma membrane. Formation of these invaginations is dependent on the expression of Caveolin-1 or -3 and proteins of the cavin family. In response to stress, caveolae disassemble and cavins are released from caveolae, allowing cavins to potentially interact with intracellular targets. Here, we describe the intracellular (non-plasma membrane) cavin interactome using biotin affinity proteomics and mass spectrometry. We validate 47 potential cavin-interactor proteins using a cell-free expression system and protein-protein binding assays. These data, together with pathway analyses, reveal unknown roles for cavin proteins in metabolism and stress signaling. We validated the interaction between one candidate interactor protein, protein phosphatase 1 alpha (PP1α), and Cavin-1 and -3 and show that UV treatment causes release of Cavin3 from caveolae allowing interaction with, and inhibition of, PP1α. This interaction increases H2AX phosphorylation to stimulate apoptosis, identifying a pro-apoptotic signaling pathway from surface caveolae to the nucleus.

[1] The University of Queensland, Institute for Molecular Bioscience, Brisbane, QLD 4072, Australia. [2] The University of Queensland Diamantina Institute, Faculty of Medicine, The University of Queensland, Translational Research Institute, Brisbane, QLD 4102, Australia. [3] The University of Queensland, Centre for Microscopy and Microanalysis, Brisbane, QLD 4072, Australia. [4]Present address: EMBL Australia Node in Single Molecule Science, Lowy Cancer Building, Level 3 Medical Sciences UNSW Kensington Campus, Sydney, NSW 2052, Australia. [5]Present address: Science and Engineering Faculty, Earth, Environmental and Biological Sciences, Genetics & Biotechnology, Queensland University of Technology, Brisbane, QLD 4000, Australia. [6]Present address: Institute for Future Environments, IFE Centres, CTCB Centre Office, Queensland University of Technology, Brisbane, QLD 4000, Australia. [7]These authors contributed equally: Kerrie-Ann McMahon, Yeping Wu. Correspondence and requests for materials should be addressed to R.G.P. (email: r.parton@imb.uq.edu.au)

Caveolae are a major membrane domain common to most vertebrate cells. Morphologically, caveolae appear as 50–100 nm vesicular structures near or attached to the plasma membrane[1]. One of the defining features of this domain is the integral membrane protein Caveolin-1 (CAV1). CAV1 is a defining structural component of caveolae that regulates diverse cellular processes, including endocytosis, vesicular transport, and mechanoprotection[1]. Recently, a family of cytosolic coat proteins, Cavin1/PTRF (caveolae-associated protein 1/Pol 1 transcription release factor), Cavin2/SDPR (caveolae-associated protein 2/serum-deprivation response protein), Cavin3/PRKCDBP (caveolae-associated protein 3/serum-deprivation response protein that binds to C-kinases), and Cavin4/MURC (caveolae-associated protein 4/Muscle-restricted coil–coil protein), were identified[2–7].

The role of caveolae in mechanoprotection was first observed when the structural integrity of caveolae was disrupted by increased mechanical tension[8,9]. Under these conditions, caveolae flatten at the plasma membrane releasing cavin coat proteins into the cytosol. We proposed that these non-caveolar cavins may propagate signals to diverse intracellular effectors and have sought to establish the protein interaction network for non-caveolar cavins.

At present, no systematic analysis of the intracellular cavin interactome has been published. Cavin1 was initially demonstrated to be involved in transcriptional regulation by interacting with RNA polymerase 1 and dissociating the paused transcription complex involving transcription termination factor 1 (TTF-1)[10]. Cavin2 was originally described as a phosphatidylserine binding protein in platelets[11]. Subsequently, rat Cavin2 was isolated as a PKC-alpha binding protein and was observed to colocalize with CAV1 at caveolae[12]. Known Cavin3 interacting proteins include Myosin 1c[13], c-Myc[14], and the period circadian protein homolog 2 (PER2) and cryptochrome circadian regulator 2 (CRY2) complex involved in circadian rhythm[15].

In order to define the intracellular cavin interactome, we utilized BioID/mass spectrometry (MS) of Cavin3. We speculated that release of cavins from caveolae would allow their interaction with intracellular targets. We therefore utilized MCF-7 cells, that lack caveolins, cavins and caveolae, as a model system to screen for putative interactors. Expressed cavin proteins in MCF-7 cells exhibit a cytosolic localization that mimics release of cavins from caveolae in cells subjected to increases in plasma membrane tension[8,9]. Using this model system, we have generated a comprehensive list of potential interacting proteins for non-caveolar Cavin3. We have complemented this approach by screening in vitro expressed potential interacting proteins using Amplified Luminescent Proximity Homogeneous Assay Screen (ALPHAScreen; refs. [16–18]) as well as GFP-Trap pulldowns of Cavin3 in both MCF-7 and A431 cells.

We now demonstrate that in cells with caveolae and with endogenous cavin proteins, cavin proteins are released from caveolae in response to cellular stressors to allow their association with endogenous PP1α validating our approach. Our results suggest that released Cavin3 from caveolae plays an important proapoptotic role in response to UV treatment through interaction with, and inhibition of, PP1α and provides the first identification of the cavin proteins as putative cytosolic signaling molecules.

## Results

**BioID analysis of non-caveolar cavin-interacting proteins.** In order to define the non-caveolar cavin interactome, we utilized the BioID proximity-based biotinylation approach that allows for the characterization of protein–protein interactions in living cells[19]. This method has the advantage of detecting weak and/or transient interactors without requiring that protein–protein interactions be maintained post lysis. As a system to identify potential non-caveolar cavin-interacting proteins, we sought a readily transfected cell line lacking caveolins and cavins in which we could identify non-caveolar and intracellular cavin-interacting proteins. For this, we tested a number of commonly used cell lines and found that MCF-7 cells (ATCC HTB-22) lack CAV1, Cavin1, Cavin2, Cavin3 at the mRNA level (Supplementary Fig. 1e) and Cavin1 and CAV1 at the protein level (Supplementary Fig. 1b–d). In the absence of CAV1 (and caveolae), expressed cavin proteins show a cytosolic distribution (Supplementary Fig. 1f–i). This allows us to mimic the interaction between cavins after release from caveolae and downstream proteins. CAV1-GFP had a Golgi/vesicular distribution in MCF-7 cells (Supplementary Fig. 1j). Furthermore, expression of Cavin1, Cavin2, Cavin3, or CAV1 alone in MCF-7 cells did not induce endogenous expression of the other cavin or CAV1 genes (Supplementary Fig. 1a–d). Cavin3 can be released from caveolae in response to increased tension on the membrane as a Cavin1–Cavin3 subcomplex or as monomeric Cavin3 which is a unique feature among all cavin proteins[8]. We therefore generated a Cavin3 construct tagged with BirA in an IRES-GFP vector to better control expression and expressed Cavin3-BirA alone in MCF-7 cells followed by streptavidin affinity purification and peptides identified by MS. Cell extracts expressing BirA alone were subjected to the same analysis as the control. Nonspecific polypeptides from cells expressing BirA alone were removed from our proteomics lists and only proteins with two or more peptides were subsequently included.

For non-caveolar Cavin3, 29 specific proteins were identified including fructose-bisphosphate aldolase (FBA) A and pyruvate kinase M (PKM), as well as the serine/threonine-protein phosphatase PP1alpha (PP1α) (yellow boxes, Supplementary Data 1). Ingenuity pathway analysis revealed several enzymes in glycolysis/gluconeogenesis for non-caveolar Cavin3 and molecular and cellular functions related to cell death and survival (Supplementary Fig. 2a).

In addition to the BioID/MS and as an independent validation, Cavin3-GFP was expressed in MCF-7 (cytosolic localization, gray boxes) and A431 cells (caveola localization, blue boxes) and GFP Trap pulldown experiments were performed followed by MS (Supplementary Data 1). Cell extracts expressing GFP alone were subjected to the same analysis and polypeptides that interact nonspecifically with the GFP beads were removed from our proteomics lists. Fifty-six specific proteins were observed in the Cavin3-GFP/MCF-7 GFP pulldowns (gray boxes, Supplementary Data 1) and 44 specific proteins were observed in the Cavin3-GFP/A431 GFP pulldowns (blue boxes, Supplementary Data 1). Five proteins interacted with Cavin3 in both MCF-7 and A431 GFP pulldowns, including histone deacetylase 2 (HDAC2), TAR DNA binding protein (TARDBP) 43, gamma-glutamylcyclotransferase (GGCT), annexin A5 (ANXA5), and PP1α (purple boxes, Supplementary Data 1).

**ALPHAScreen/in vitro expression analysis.** To begin characterizing these potential non-caveolar cavin-interacting proteins, 47 proteins were chosen for further analysis using ALPHAScreen, a sensitive bead-based proximity assay[8,16–18,20]. The protein list was a broad selection of proteins with diverse functions derived from the BioID, such as peroxiredoxin-6 (PRDX6) and the PP1α, and from literature searches as either potential cavin-interacting proteins (Supplementary Table 1) or proteins with links to caveolar functions. For example, dual specificity tyrosine-(Y)-phosphorylation regulated kinase 3 (DYRK3) and serine/threonine-protein kinase A-Raf (ARAF) were identified as proteins important in caveolar-mediated endocytosis[21]. The circadian

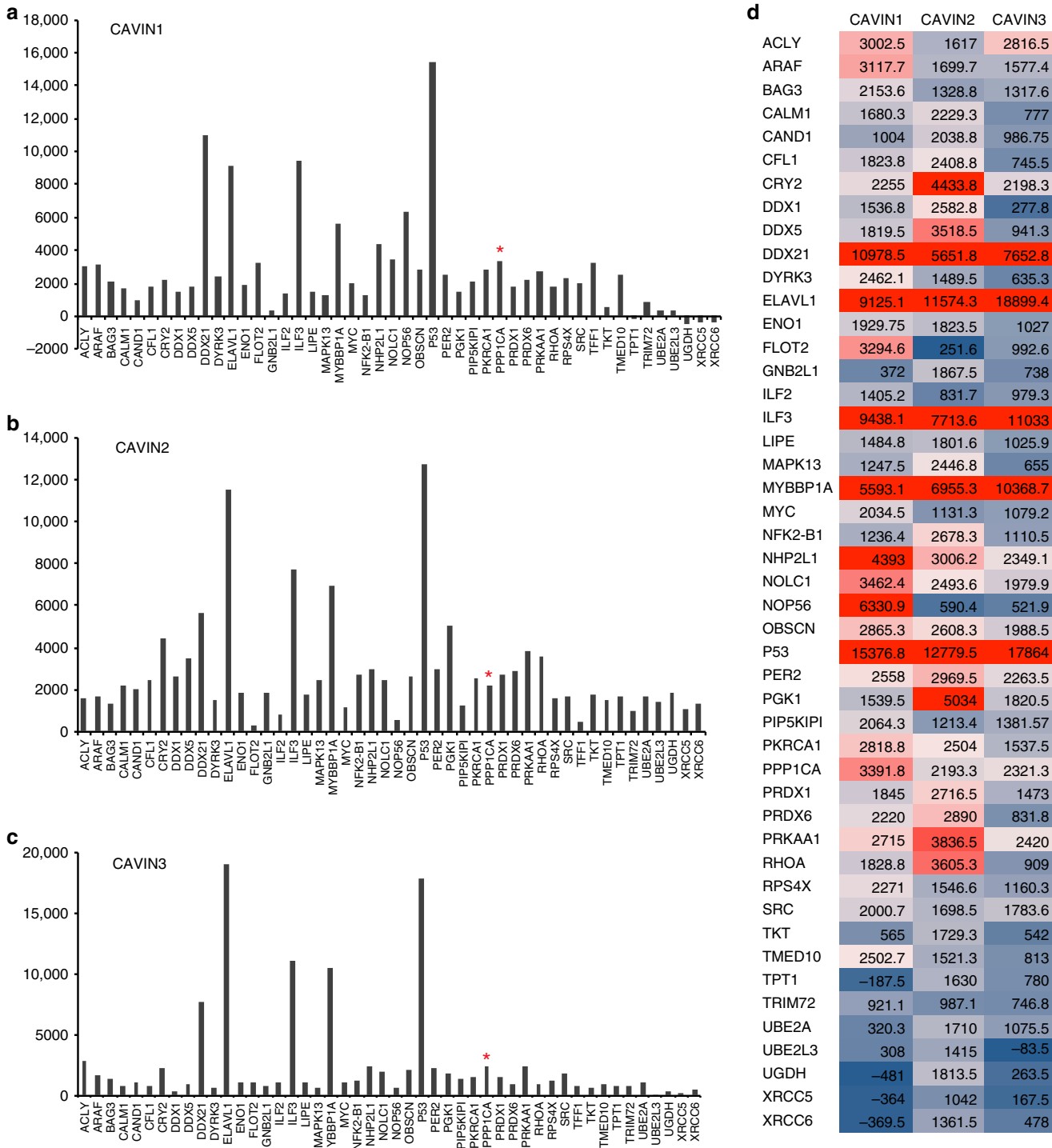

**Fig. 1** ALPHAScreen analysis of interacting proteins. **a–c** Cavin1 (**a**), Cavin2 (**b**), and Cavin3 (**c**) following co-expression of 47 potential interacting proteins in the eukaryotic *Leishmania tarentolae*-based cell-free-expression system. Each protein pair was tested in triplicate and values above 2000 counts were considered positive interactions. Red = positive interactions, blue = no interactions. *PPP1CA serine/threonine-protein phosphatase PP1alpha catalytic subunit. **d** Interaction heatmap of the mean values of 47 tested potential interacting proteins with each of Cavin1, Cavin2, and Cavin3 from three independent experiments

rhythm proteins, PER2, and CRY2 were previously identified in a complex with Cavin1 and Cavin3[15], and Cavin1 was independently identified as a potential CRY2 interacting protein[22]. ATP citrate lyase (ACLY) exhibited decreased expression in adipose tissue of Cavin3 knockout mice and was also included in the ALPHAScreen[23]. We next systematically analyzed pairwise direct interactions between each of the cavin proteins, Cavin1 (Fig. 1a),

Cavin2 (Fig. 1b), and Cavin3 (Fig. 1c), following co-expression of 47 possible interacting partners in our cell-free system. Cavin2 was included here for comparison. The recently developed eukaryotic *Leishmania tarentolae*-based cell-free-expression system (LTE) was used to accelerate conversion of DNA to protein[24,25] and allows examination of direct protein–protein interactions with expression of full-length, functional proteins

with precise stoichiometry control of the co-expressed proteins in a 3 h period[16–18,24,25]. We recently demonstrated that cavin proteins produced in this system have the same behavior as in mammalian cells[8]. In the interaction screen, each protein pair was reconstituted by co-expression, and relative expression levels were determined by measuring the fluorescence of GFP and mCherry tags. From previous experiments[16,17], a threshold of above 2000 (background) selected positive interactions and was employed in these studies. For Cavin3, this threshold corresponds to the top 20–25% binding intensities. The luminescence intensities are plotted as averages over three independent expressions/experiments where for each experiment, the signal intensity from a negative control, GFP alone in solution, was subtracted. An average of all the normalized data for both configuration of each protein pair (GFP-protein A/protein B-Cherry or GFP-protein B/ protein A-Cherry) was calculated to provide the binding index that is presented using a heatmap plot (Fig. 1d). Positive interactions are indicated as red bars. In addition, ALPHAScreen intensity response as a function of protein dilution was measured for different proteins binding to Cavin3 (Supplementary Fig. 3a) as well as for each of the cavin proteins and PP1α (Supplementary Fig. 3b).

A proportionate Venn diagram was generated to determine the interacting proteins that are common and specific for each of Cavin1, Cavin2, and Cavin3 from the ALPHAScreen (Supplementary Fig. 4). This analysis revealed 13 proteins, including ELAV-like RNA-binding protein 1 (ELAV1), p53, and PP1α, that can interact independently with each cavin proteins. ACLY specifically interacted with Cavin1 and Cavin3, while nucleolar protein 56 (NOP56) interacted only with Cavin1. Taken together these complementary proteomic and interaction approaches provided a tentative list of potential non-caveolar intracellular cavin-interacting proteins and helped establish bona fide interactors for further analysis (Supplementary Data 1). These were derived from BioID/MS (yellow, Supplementary Data 1), ALPHAScreen (green, Supplementary Data 1), literature searches (orange, Supplementary Data 1), Cavin3-GFP/MCF-7/MS (gray, Supplementary Data 1) and Cavin3-GFP/A431/MS (light blue, Supplementary Data 1), and proteins identified in both GFP Trap MCF-7 and A431 cells (purple boxes, Supplementary Data 1). The approaches used (ALPHAScreen, BioID and GFP Trap/MS) have differing advantages/disadvantages for protein identification (see Discussion section). Given that PP1α was the only protein identified in multiple approaches (BioID and GFP Trap/MS experiments from A431 and MCF-7 cells expressing Cavin3-GFP), and importantly, showed a direct interaction with in vitro synthesized cavins by ALPHAScreen, we focused our attention on the role of this interaction in our model cell systems.

**PP1α associates with cavin proteins**. In order to validate the interaction between PP1α and the cavin proteins, we first exogenously expressed mCherry-tagged Cavin1, Cavin2, and Cavin3, respectively, with GFP-PP1α in MCF-7 cells. Western blot analysis of the cytosolic fractions after GFP Trap pulldown revealed coprecipitation of mCherry-Cavin1 and mCherry-Cavin3 with GFP-PP1α (Fig. 2a) consistent with the interaction profile from the ALPHAScreen analysis. Conversely, western blot analysis of GFP Trap pulldown from MCF-7 cell cytosolic fractions with exogenously expressed GFP-tagged Cavin1, Cavin2, and Cavin3 coprecipitated with endogenous PP1α (Fig. 2b). Cavin2-GFP exhibited lower interaction levels with endogenous PP1α compared with Cavin1-GFP and Cavin3-GFP (Fig. 1b), which is consistent with the results shown in the cell-free system by ALPHAScreen (Fig. 1d). These results collectively suggest a weaker or less stable interaction between PP1α and Cavin2,

therefore, all subsequent experiments focused on the interaction of PP1α with Cavin1 or Cavin3.

As an independent test of association of cavin proteins with PP1α in cells, proximity ligation assay (PLA) was performed to assess the association of endogenous PP1α with exogenously expressed GFP-tagged Cavin1 and Cavin3. As shown in Fig. 2c, d, there was a significant association between each of the GFP-tagged cavin proteins with PP1α as compared with control GFP/ MCF-7 cells. Single-molecule coincidence (SMC) analysis[8] was further employed to validate the interaction between PP1α and Cavin1 and Cavin3. Based on the larger amplitude of the bursts observed for Cavin1, we can deduce that Cavin1 primarily exists in an oligomeric state when expressed in MCF-7 cells. The amplitude of bursts observed for Cavin3 correspond to the brightness of individual mCherry fluorophores, confirming that Cavin3 primarily exists in a monomeric state (Fig. 2e, f). The coincidence ratio for GFP-PP1α and mCherry-Cavin1 at 0.7 suggested that PP1α binds to oligomeric Cavin1. For Cavin3, the coincidence ratio at 0.5, suggested a 1:1 interaction between PP1α and Cavin3 (Fig. 2f). These findings confirm that heterologous expressed Cavin1 and Cavin3 can associate with PP1α in these model systems.

We further characterized the interaction between Cavin3 or Cavin1, and PP1α. We designed truncation mutants guided by amino acid alignment to the two major domains in Cavin1 and Cavin3, helical region 1 (HR1; amino acids aas 48–164 in Cavin1; aas 14–130 in Cavin3; Fig. 3a) and HR2 domain (aas 209–300 in Cavin1; aas 160–210 in Cavin3; Fig. 3a)[26]. AlphaLISA assay was used to test the interaction between in vitro synthesized PP1α and truncation mutants of Cavin3 and Cavin1. The co-expression efficiency of PP1α-mCherry and GFP-tagged cavin constructs were examined in an LTE system (Supplementary Fig. 5). As was observed in MCF-7 cells, constructs containing the HR2 domain of Cavin3 can associate with PP1α but not those containing only the HR1 domain (Fig. 3b). These results were further validated by GFP-Trap assay in MCF-7 cells transfected with GFP-tagged transgenes. Endogenous PP1α interacted with full-length Cavin3, a mutant containing both the HR1 and HR2 domains, the HR2 domain alone, but not the HR1 domain alone (Fig. 3d, e). In addition, the AlphaLISA assay revealed that, similar to Cavin3, constructs containing the HR2 domain of Cavin1 showed significantly increased association with PP1α compared with the HR1 domain alone. Collectively, these findings suggest that PP1α interacts directly with the HR2 domains of Cavin1 and Cavin3.

**PP1α associates with cavin proteins released from caveolae**. We next sought to investigate the proposed interaction between PP1α and cavin proteins in a cellular system. We focused on A431 cells that have been used extensively for caveolae studies due to an abundance of caveolae[27,28] and because, as an epidermoid skin cancer cell line, these cells have been used extensively in experiments investigating UVC irradiation, as explored later in this study[29,30]. During the course of these experiments, we determined that A431 cells (similar to several other commonly cancer cell lines such as HeLa cells) do not express Cavin2 but have significant expression of the universal components of caveolae, Cavin1, CAV1, and Cavin3. We also employed the human breast cancer cell line, MDA-MB-231 as an additional cell type.

Under control conditions PP1α was mainly localized in the nucleus of A431 cells while the Cavin1 and Cavin3 proteins exhibited a predominantly caveolar distribution at the plasma membrane (Fig. 4a, b, upper panels). This lack of colocalization suggests that the proteins can only interact if caveolae are disassembled. As a model system we subjected A431 cells to hypo-osmotic medium to induce caveola disassembly. Electron

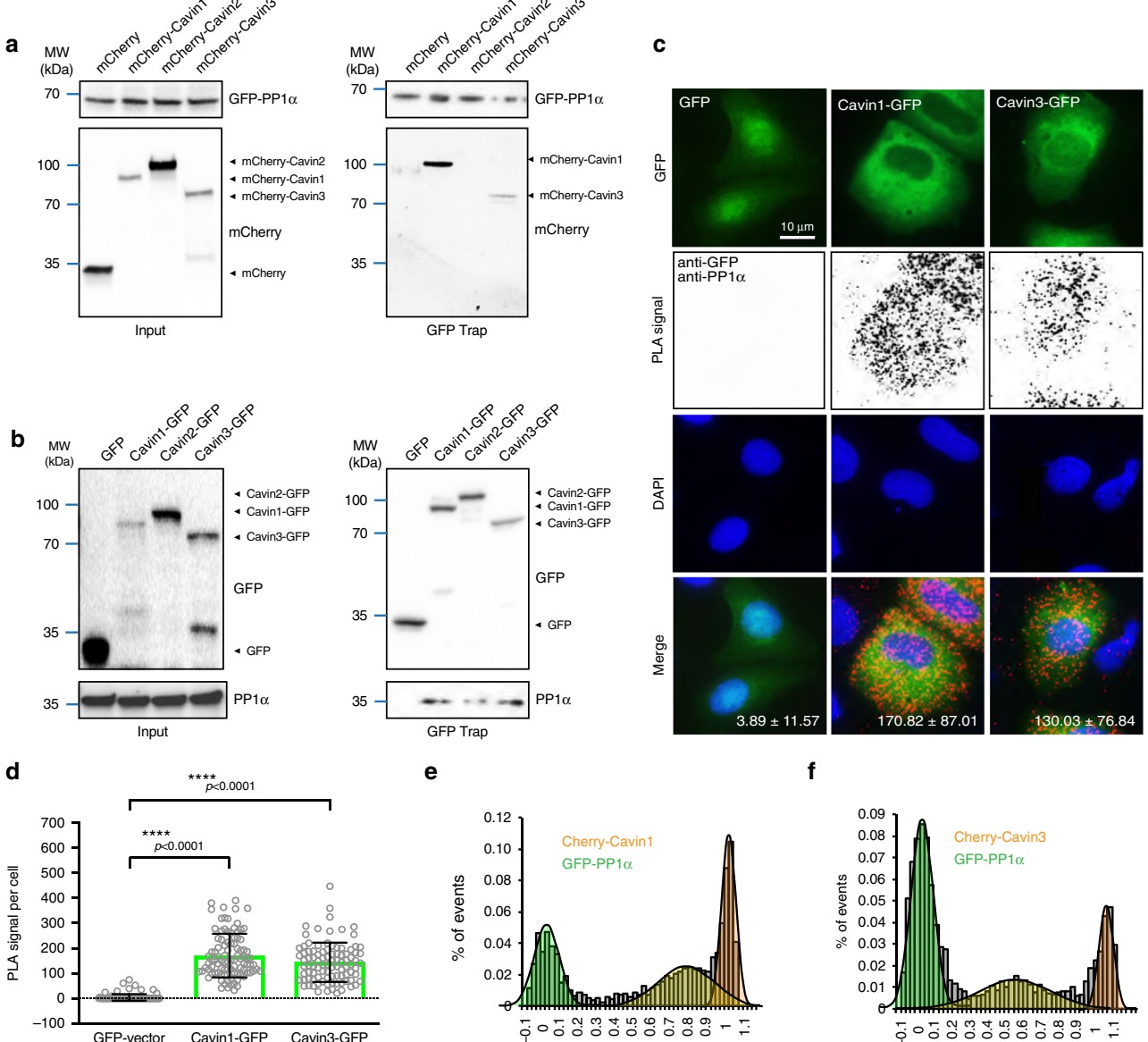

**Fig. 2** Interaction between the cavin proteins and PP1α. **a** GFP Trap assays of MCF-7 cells cotransfected with GFP-PP1α and mCherry-vector, mCherry-Cavin-1, -2 or -3. Transfection efficiency was examined in the input material (left panel). mCherry was detected in the GFP pulldown samples by western blot analysis (right panel). **b** Input and pulldown samples for GFP Trap assays of Cavin-GFP cells were analyzed by western blotting for GFP and PP1α, with GFP-vector as the negative control. **c** In situ PLA detection (red puncta) of the interaction between GFP-cavins and endogenous PP1α in MCF-7 cells using anti-GFP (mouse) and anti-PP1α (rabbit) antibodies. DNA was stained by DAPI (blue). Red puncta per cell is represented as mean ± SEM for three experiments from different fields of view. Scale bar, 10 μm. **d** Quantification of PLA puncta/cell for 150 total cells ($n = 50$ cells per experiment, three independent experiments) from different fields of view. Black colored lines in the scatter plot represent the mean ± SD, ****$p$-value (one-way ANOVA) < 0.0001 versus negative control (GFP-vector/PP1α). **e-f** Single-molecule coincidence detection between GFP-PP1α and mCherry-Cavin1 (**e**) or mCherry-Cavin3 (**f**). Coincidence ratio value ($C$) = intensity (A549 nm)/total intensity

microscopy revealed a loss of morphological caveolae (caveolae per length of plasma membrane of cells incubated in hypo-osmotic medium, 65.5 ± 8% of iso-osmotic treated A431 cells; Supplementary Fig. 6). We then used PLA to assess interaction of endogenous Cavin1 and Cavin3 and PP1α (Fig. 4c). The interaction between Cavin1 and Cavin3 was used as a positive control, with fluorescent puncta indicating interaction observed along the plasma membrane of these cells. In contrast, endogenous cavins and PP1α were not in close proximity and few PLA puncta were observed in untreated cells (Fig. 4c–e, upper panels; see PLA controls Supplementary Fig. 8), indicating few associations between PP1α and Cavin1 or Cavin3 under steady state in A431 cells. Upon hypo-osmotic treatment, there was a

dramatic increase in association of endogenous PP1α, and Cavin1 or Cavin3 (a fourfold increase in Cavin1-PP1α and a 2.6-fold increase in Cavin3-PP1α interactions; Fig. 4d and f; Fig. 4e and h). The ratio of nuclear and cytosolic localized interaction signals was calculated and revealed that the majority of Cavin1-PP1α (78.6 ± 6.9%; Fig. 4g) and Cavin3-PP1α (73.9 ± 3.3%; Fig. 4i) complexes formed after hypo-osmotic treatment were localized in the cytosol. Consistent with the quantitative data, the redistribution of PP1α from the nucleus to the cytosol, as well as the release of both Cavin1 and Cavin3 from plasma membrane caveolae to the cytosol, was observed by immunofluorescence (Fig. 4a, b, lower panels). As in A431 cells, the redistribution of PP1α into the cytosol still occurred in cavin-deficient MCF-7 cells following

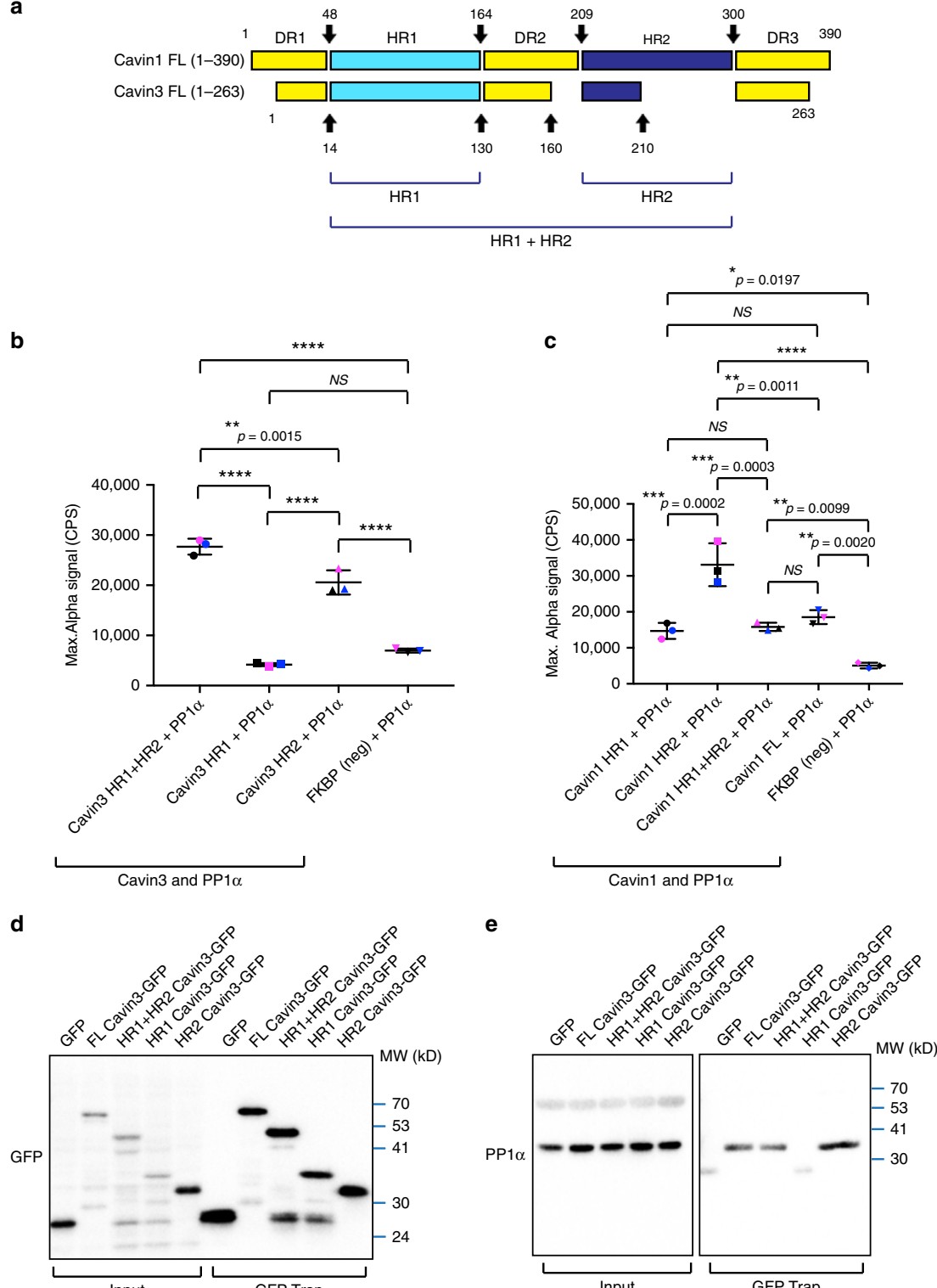

**Fig. 3** PP1α interacts with the HR2 domain of Cavin3 and Cavin1. **a** Schematic representation of the helical regions (HR)1 and HR2 regions of Cavin1 and Cavin3. **b** AlphaLISA screen of Cavin3 HR1 domain, Cavin3 HR2 domain, and Cavin3 HR1+HR2 domain and PP1α, FKBP, and PP1α (negative control), **$p$-value < 0.01, ****$p$-value < 0.0001 (one-way ANOVA). NS = no significance. Scatter dots in each group represent the values from three independent experiments (labeled by three colors). **c** AlphaLISA screen of Cavin1 HR1 domain and PP1α, Cavin1 HR2 domain and PP1α, Cavin1 HR1+HR2 domain and PP1α, Cavin1 FL and PP1α, and FKBP and PP1α (negative control), *$p$-value < 0.05, **$p$-value < 0.01, ***$p$-value < 0.001, ****$p$-value < 0.0001 (one-way ANOVA). Mean ± SD are presented for three sets of experiments. **d–e** GFP-tagged full-length Cavin3 (lane 2) and truncation mutants (HR1+HR2) domain, HR1 domain or HR2 domain were expressed in MCF-7 cells followed by lyzed GFP Trap pulldown assays. Transfection and pulldown efficiency were examined by western blot analysis using anti-GFP antibodies (**d**). Endogenous PP1α (**e**) was detected by western blotting using PP1α specific antibodies. Results are representative of three independent experiments

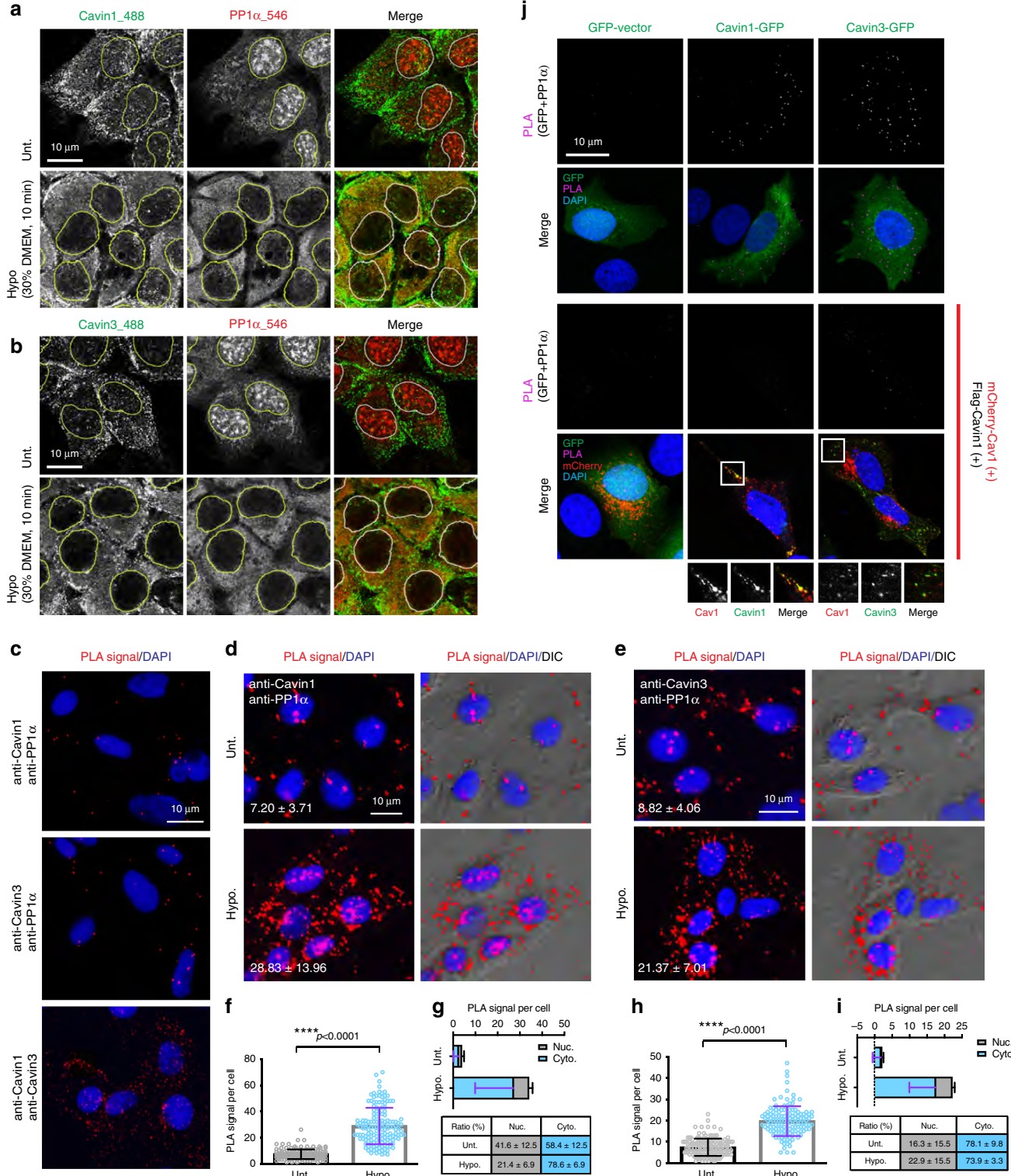

hypo-osmotic treatment, suggesting that hypo-osmotic stress-induced translocation of PP1α is a cavin/caveolae-independent process (Supplementary Fig. 7).

These results strongly suggest that the release of cavins is required for association with PP1α. To further test this, we expressed GFP-tagged Cavin1 and 3 with PP1α in MCF-7 cells with or without the co-expressed caveola forming proteins, CAV1 and Cavin1 (Fig. 4j). Cavin1-GFP or Cavin3-GFP alone showed cytosolic distribution (Fig. 4j, upper two panels).

With the addition of Flag-Cavin1 (except for Cavin1-GFP expressing cells) and mCherry-CAV1 (Fig. 4j, lower two panels), Cavin1-GFP and Cavin3-GFP were recruited to caveolae, whereas a negative control GFP-vector remained in the cytosol. PLA signals for the association of PP1α and GFP-tagged Cavin1 or Cavin3 were downregulated in caveola-rescued MCF-7 cells (Fig. 4j, lower two panels), suggesting that the sequestration of cavins at caveolae abolishes their interaction with PP1α.

**Fig. 4** Hypo-osmotic interactions of endogenous PP1α and cavin proteins. **a**, **b** Representative immunofluorescence images of endogenous Cavin1 (green) and PP1α (red) (**a**) or Cavin3 (green) and PP1α (red) (**b**) in untreated or hypo-osmotic treated A431 cells. Yellow or white circles (in merged images) indicate the outline of the nucleus. Scale bar, 10 μm. Images are representative of three independent experiments. **c** Representative images of PLA signals (red)/cell representing the interactions of endogenous PP1α/Cavin1 or PP1α/Cavin3 in A431 cells from three independent experiments. Cavin1/Cavin3 interaction was used as a positive control. DAPI was used to visualize nuclear DNA (blue). Scale bar, 10 μm. **d**, **e** PLA signal/cell of Cavin1/PP1α and Cavin3/PP1α interactions in A431 cells with or without hypo-osmotic treatment. The PLA signals between cells were distinguished by the matched DIC microscopy and are represented as the mean ± SD. Scale bar, 10 μm. **f**, **h** Quantitation of PLA signals for the Cavin1/PP1α (**f**) and Cavin3/PP1α (**h**) interactions upon hypotonic treatment, $n = 150$ cells for three independent experiments. ****$p$-value (Student's $t$-test, two tail) < 0.0001. **g**, **i** The number and ratio (%) of nuclear (Nuc.) and cytoplasmic (Cyto.) localized PLA signal for Cavin1/PP1α (**g**) and Cavin3/PP1α (**i**) interactions was quantified using CellProfiler, where DAPI staining was utilized for nuclear identification. Histogram represents the mean ± SD values of PLA signal number in each cell from three independent experiments ($n = 50$ cells for each experiment). Cytoplasmic and nuclear signals were labeled as blue and gray. The values of the ratio are presented in in the table as mean ± SD. **j** MCF-7 cells transfected with GFP-cavins and GFP-vector (upper two panels), or with additional mCherry-CAV1 and Flag-Cavin1 (not expressed in Cavin1-GFP cells) (lower two panels). PLA signals for the interaction between GFP and PP1α were detected and visualized as magenta puncta in merged images. PLA signal images alone were inverted to gray scale. Enlarged images showing the colocalization between mCherry-CAV1 and GFP-tagged Cavin1 and Cavin3 are presented. Scale bar, 10 μm. Images are representative of three independent experiments

**Cavin3 and PP1α interact in response to UV stress**. Having established a sensitive assay for cavin-PP1α association, we next asked whether other stimuli could similarly induce caveolar disassembly and facilitate the interaction between Cavin3 and PP1α. We focused on Cavin3 and its response to UV since its expression has been demonstrated to increase cellular sensitivity of cells to various stressors[31,32]. We first treated A431 cells with UV (2 min treatment and further 4 h incubation) that induced significant redistribution of Cavin3 from the plasma membrane to the cytosol and nucleus and colocalization with PP1α (Supplementary Fig. 10a). We next used PLA to detect co-association of endogenous Cavin3 and PP1α. As shown in Fig. 5a (quantification in Fig. 5b, also see PLA controls in Supplementary Fig. 8c), a significant increase in the interaction of PP1α and Cavin3 after UV treatment (Fig. 5b) was observed mainly in the nucleus (66.1 ± 2.0%; Fig. 5c) of these cells. Moreover, an increase in the interaction of Cavin3 with PP1α after UV treatment of A431 cells was also detected using a GFP Trap pulldown approach (Supplementary Fig. 10b). Consistent with a model in which caveolar disassembly is required for cavin release from caveolae, morphologically recognizable caveolae were consistently reduced in UV-treated cells (density of apical caveolae 60 ± 11% (SEM) of control cells; Fig. 5d).

To test whether other cell types show a similar UV-mediated redistribution of Cavin3 to allow interaction with PP1α, we tested the effect of UV treatment on the MDA-MB-231 cells that demonstrated a similar increased interaction between Cavin3 and PP1α in both the cytosol and nucleus following UV treatment (Supplementary Fig. 9). Collectively, these results suggest that the redistribution of Cavin3 and interaction with PP1α is a general response to UV-induced stress.

**Cavin3 interacts with PP1α to regulate UV apoptosis**. We next hypothesized that the induced interaction between Cavin3 and PP1α might influence a UV-induced downstream cellular response. UV stress causes DNA damage and drives signaling pathways that determine cell fate, DNA damage-induced cell death or survival[33,34]. We depleted A431 cells of PP1α or Cavin3, respectively, using an siRNA-approach and measured the activation of DNA damage response proteins. We first assessed the phosphorylation of H2AX, termed γH2AX, as a DNA damage marker[35,36]. γH2AX levels were increased in PP1α-depleted A431 cells (Fig. 5e, quantification in Fig. 5f, panel i). In contrast, γH2AX levels were significantly reduced in Cavin3-depleted A431 cells (Fig. 5e, quantification in 5f, panel i). These results suggested that Cavin3 and PP1α play opposite roles in DNA damage signaling as measured by H2AX phosphorylation. Furthermore, in the absence of PP1α, Cavin3 depletion (cells with

PP1α-siRNA and Cavin3-siRNA knockdown) showed no significant effect on the generation of γH2AX following UV irradiation compared with control cells (scrambled siRNA) (Fig. 5e, quantification in 5f, panel i), indicating that Cavin3-mediated UV responsive regulation on γH2AX is PP1α dependent. These findings were further confirmed in reciprocal PP1α and Cavin3 overexpression experiments (Supplementary Fig. 10c, quantification in Supplementary Fig. 10d).

These findings led us to directly test whether the Cavin3-PP1α system regulates apoptosis. We immunoblotted for cleaved-caspase-3 and cleaved PARP, classical apoptosis markers (Fig. 5g). Similar to γH2AX protein levels, cleaved-caspase-3 and PARP were both decreased in Cavin3-depleted A431 cells and upregulated in PP1α-depleted A431 cells after UV treatment compared with control treated cells (Fig. 5g, quantification in Fig. 5h). In addition, these apoptotic markers showed no significant difference in cells with double knockdown of PP1α and Cavin3 compared with control cells (scrambled siRNA) (Fig. 5g, quantification in 5h). Reciprocal results were observed in A431 cells overexpressing Cavin3 or PP1α, respectively (Supplementary Fig. 10e, quantification in Supplementary Fig. 10f). These data further suggested that PP1α is a major intracellular target for Cavin3 in the regulation of UV-induced apoptosis. As an independent validation of the western analysis, we measured LDH release as an index of plasma membrane damage in cells after UV treatment. LDH release after UV treatment was increased in PP1α-depleted A431 cells and decreased in Cavin3-depleted A431 cells (Fig. 5i). Simultaneous knockdown of both PP1α and Cavin3 in A431 cells slightly neutralized the effect of PP1α depletion on LDH release in these cells (Fig. 5h). These findings were further confirmed in reciprocal overexpression experiments (Supplementary Fig. 10g). Simultaneous overexpression of both PP1α and Cavin3 in A431 cells restored LDH release after UV treatment to control levels (Supplementary Fig. 10g). Collectively, these data suggest that interaction with PP1α is crucial for released Cavin3 to promote apoptosis following UV treatment.

**Cavin3 attenuates PP1α activity and promotes H2AX phosphorylation**. PP1α modulates signaling pathways by dephosphorylating its substrates[37–39]. Our experiments demonstrated that PP1α negatively regulates the phosphorylation of H2AX in response to UV-induced DNA damage raising the possibility that γH2AX, may be a substrate of PP1α.

To test this, we first examined the levels of γH2AX levels in A431 cells treated with tautomycin (TTM), a selective inhibitor of PP1α[40]. As shown in Fig. 6a, the generation of γH2AX upon UV irradiation was significantly upregulated in TTM treated cells

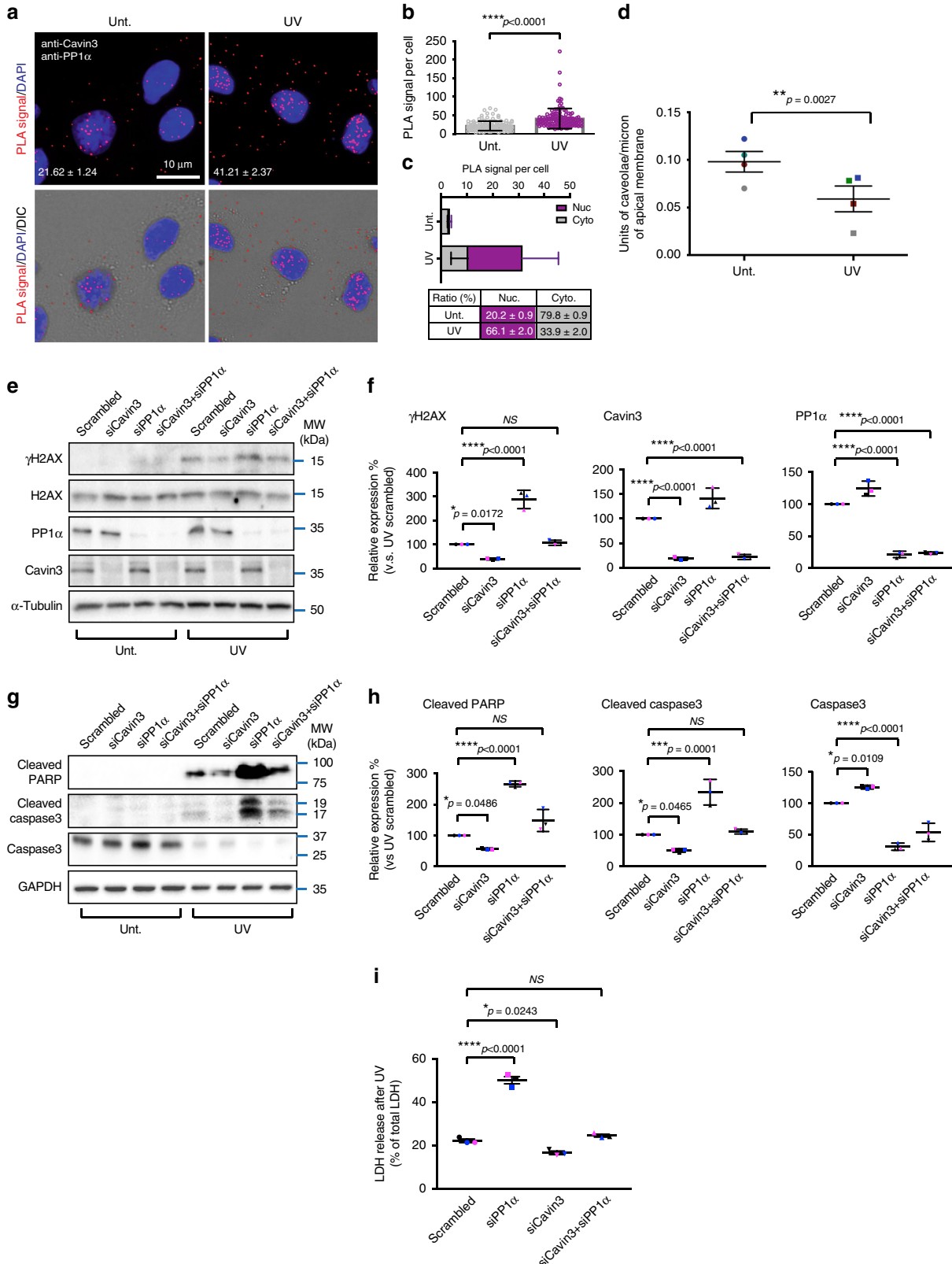

(quantification in Fig. 6b), suggesting that PP1α phosphatase activity contributes to the dephosphorylation of γH2AX upon UV treatment. We further examined whether Cavin3 and PP1α form a complex with γH2AX using a GFP Trap pulldown in control or UV-treated A431 cells transfected with PP1α-GFP or Cavin3-GFP. γH2AX was specifically coprecipitated by either PP1α-GFP or Cavin3-GFP only in response to UV treatment suggesting the formation of a UV-induced complex between these proteins (Fig. 6c, d). PLA further confirmed a predominant nuclear association of endogenous γH2AX and PP1α (82.3 ± 8.6%; Fig. 6e, f), as well as γH2AX Cavin3 (83.5 ± 1.5%; Fig. 6g, h), specifically after UV treatment.

**Fig. 5** PP1α-Cavin3 complex in UV-induced DNA damage response. **a** PLA interaction between Cavin3 and PP1α upon UV treatment. DAPI detects the nucleus (blue). **b** Quantification of PLA Cavin3/PP1α interactions in control and UV-treated cells as the mean ± SD, three individual experiments. ****$p$-value (Student's $t$-test, two tail) < 0.0001. **c** The number and ratio of nuclear (Nuc.; purple) and cytoplasmic (Cyto.; gray) localized PLA signals for Cavin3-PP1α interactions presented as mean ± SD, three independent experiments in a histogram and table. **d** Quantitation of caveolae number in untreated and UV-treated A431 cells. Caveolae number on the apical surface from four sets of images were quantified blinded from three independent experiments and were normalized to the length of the surface sampled (over 150 μm for each set of images where **$p = 0.0027$, Student's $t$-test, two tail). **e** A431 cells transfected with scrambled siRNA (control), Cavin3 and PP1α individually or together were exposed to UV treatment. Lysates were western blotted for γH2AX and H2AX anti-PP1α and anti-Cavin3 and α-Tubulin antibodies. **f** Relative protein expression (%) of γH2AX (Scrambled vs siCavin3: *$p = 0.0172$, Scrambled vs siPP1: ****$p < 0.0001$), Cavin3 (Scrambled vs siCavin3: ****$p < 0.0001$, Scrambled vs siCavin3+siPP1: ****$p < 0.0001$), and PP1α (Scrambled vs siPP1: ****$p < 0.0001$, Scrambled vs siCavin3+siPP1: ****$p < 0.0001$), three independent experiments, one-way ANOVA with a Tukey's multiple comparison test. **g** Western blot analysis of cleaved PARP, cleaved-caspase-3, total caspase-3 and GAPDH in PP1α- or/and Cavin3-knockdown A431 cells. **h** Relative protein expression (%) of cleaved PARP (Scrambled vs siCavin3: *$p = 0.0486$, Scrambled vs siPP1: ****$p<0.0001$), cleaved caspase-3 (Scrambled vs siCavin3: *$p = 0.0465$, Scrambled vs siPP1: ***$p = 0.0001$) and total caspase-3 (Scrambled vs siCavin3: *$p = 0.0109$, Scrambled vs siPP1: ****$p < 0.0001$). Three independent experiments analyzed using one-way ANOVA. **i** LDH release (% of total LDH) was calculated (Scrambled vs siPP1: ****$p < 0.0001$, Scrambled vs siCavin3: *$p = 0.0243$). Significance was determined by one-way ANOVA and a Bonferroni test from three independent experiments

Taken together, our results suggest that Cavin3 and PP1α form a complex (Fig. 5a–c; Supplementary Fig. 8b) and exhibit opposite effects on UV-induced H2AX phosphorylation (Fig. 5e–f; Supplementary Fig. 10c, d). These results suggest Cavin3, as a PP1α interacting protein, may possess regulatory effects on PP1α that impact H2AX phosphorylation. To test this idea, the effect of Cavin3 on the phosphatase activity of PP1α was assessed using a phosphatase assay. PP1α was immunoprecipitated from MCF-7 cells with or without Cavin3 re-expression to exclude the impact of other phosphatases. Overexpression of mScarlet-PP1α, a positive control, showed significantly increased phosphatase activity compared with MCF-7 cells with endogenous PP1α expression (Fig. 6i). Cavin3 re-expression in MCF-7 cells significantly downregulated the phosphatase activity of both endogenous and overexpressed PP1α (Fig. 6i), suggesting Cavin3 negatively regulates PP1α activity. Collectively, our results suggest that released Cavin3 from caveolae plays an important proapoptotic role in UV treatment through interaction with and inhibition of PP1α. Considering the opposing roles of Cavin3 and PP1α in apoptosis and the functions of PP1α in facilitating DNA repair, we propose that the Cavin3-PP1α complex affects UV-induced DNA damage signaling by directly interacting with γH2AX and by competitively modulating the phosphorylation level of γH2AX (Fig. 7).

## Discussion

This report describes a number of complementary approaches; BioID, GFP Trap/MS, ALPHAScreen and PLA as effective methods to screen for proximate and interacting partners of intracellular localized cavin proteins. The power of our experimental approach is the identification of potential interacting proteins by heterologous expression in a cell line lacking caveolar proteins and in vitro ALPHAScreen in addition to detailed studies to validate the interaction of cavins and PP1α. These approaches (ALPHAScreen, BioID and GFP Trap/MS) have differing advantages/disadvantages for protein identification. GFP Trap pulldowns identified cavin complexes but may miss low affinity or transient interactions, whereas BioID requires biotinylation in a correct spatial proximity (estimated to be around 10 nm) with the advantage of labeling physiological interactions in live cells even when these are very weak or transient in nature[41]. In addition, ALPHAScreen in combination with in vitro synthesized cavins, is a sensitive, high throughput, systemic way to evaluate direct protein–protein interactions.

Our results show that cellular stress can cause release of cavins from plasma membrane caveolae suggesting that cavin proteins act as intracellular signaling molecules. The results of this study have general significance for a mechanistic understanding of caveola function. Caveolae are the most abundant surface organelle of many human cells and yet we still have no clear understanding of how caveolae function in healthy cells and the implications of caveolar dysfunction in numerous diseases with which caveolae have been linked. Our working hypothesis is that cavin proteins may be the key to understanding the multiple cellular pathways linked to caveolae. Previous studies have demonstrated that caveolae can be disassembled to release cavin proteins as distinct subcomplexes into the cytosol comprising Cavin1/Cavin2, Cavin1/Cavin3, and monomeric Cavin3[8,9] in response to increased membrane tension and independently, Cavin1 can be released from surface caveolae in response to insulin treatment of adipocytes allowing interaction with nuclear targets[42]. Our data now show that caveolae can also release cavins in response to "non-mechanical" stressors such as UV and can be considered as more general stress sensors. In addition, our previous studies have uncovered a pathway that maintains low cytosolic cavin protein levels under steady state conditions, a prerequisite for a signaling pathway dependent on cytosolic cavin proteins[43].

Furthermore, a number of different intracellular locations for the cavins (including association with the nucleus) have been described by us and others[42,44] which may indicate that these proteins interact with other intracellular compartments when released from caveolae. This is supported by the majority of proteins identified as potential non-caveolar cavin-interacting proteins being localized predominantly in the cytosol and nucleus that may be relevant to disease conditions. Cavin proteins are strongly linked to cancer[45–48] and it will be important to determine if the links to stress[48] and to metabolism[13] are related to changes in cavin expression and localization in cancer cells (discussed further in Supplementary Information section).

Using our interaction analyzes, we identify PP1α as an intracellular target for Cavin3 when released from caveolae in response to UV stress. We propose that the interaction between Cavin3 and PP1α influences UV-mediated apoptosis through the formation of a complex with γH2AX, where PP1α acts to inhibit γH2AX phosphorylation resulting in a reduction in apoptosis while Cavin3 acts to partially counteract PP1α functions during this nuclear event. Interestingly, Cavin3-PP1α-γH2AX complexes appear more concentrated in the nucleus (Fig. 6e, f) than Cavin3-PP1α complexes after UV treatment (Fig. 5a). This suggests that the formation of Cavin3-PP1α-γH2AX complex could be a dynamic process, whereby the released Cavin3 may interact with the cytosolic pool of PP1α first and subsequently translocate into

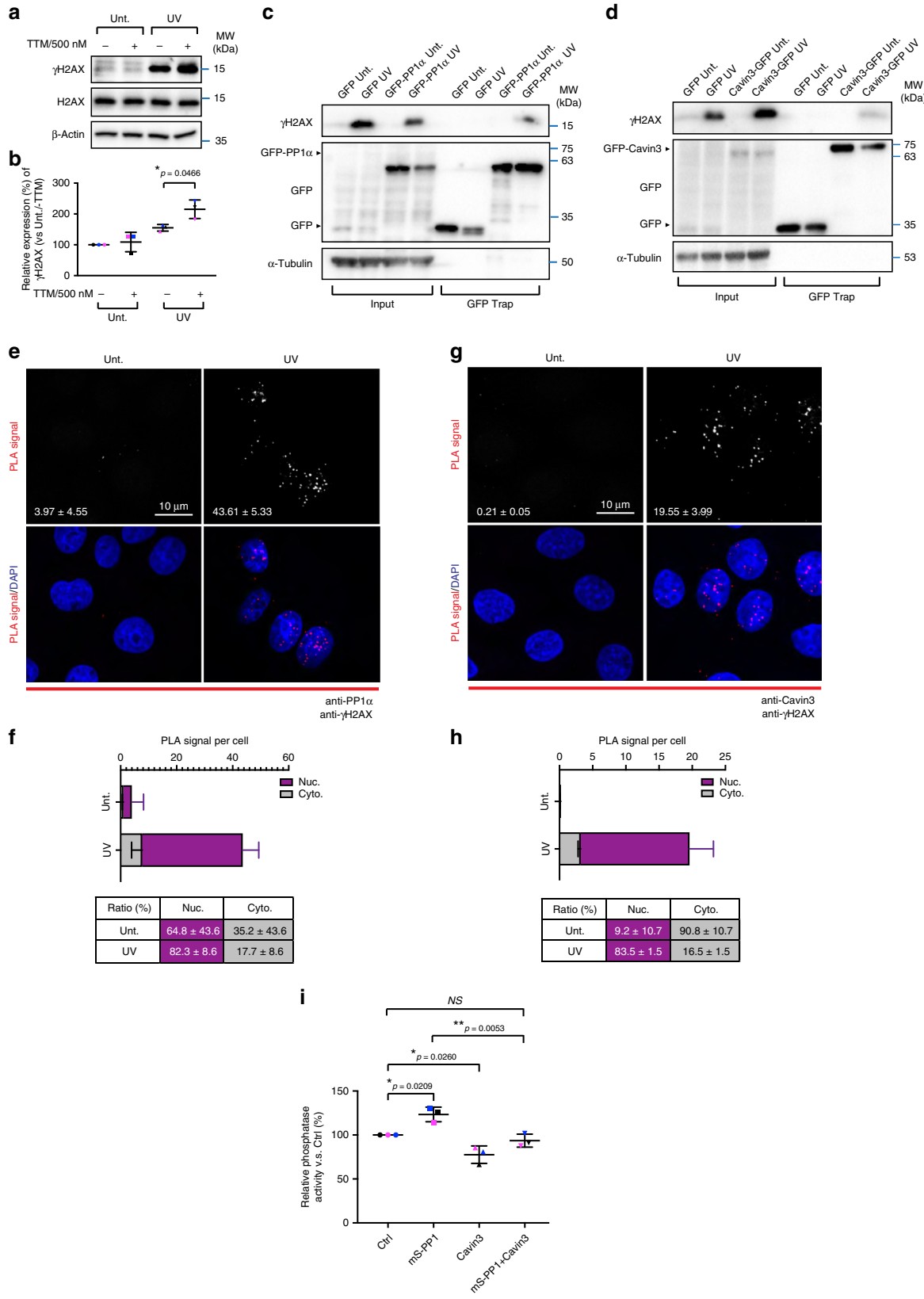

the nucleus as a Cavin3-PP1α complex to associate with γH2AX (Fig. 7). Future experimentation will be required to elucidate the precise sequence of events leading to the formation of the cytoplasmic Cavin3-PP1α complex and the nuclear Cavin3-PP1α-γH2AX complex. However, we believe that the differential distribution of Cavin3, PP1α or Cavin3-PP1α complexes under hypo-osmotic treatment and UV stress is independent of γH2AX as γH2AX is also induced in the nucleus in response to hypo-osmotic treatment[49] yet Cavin3-PP1α complex is mainly cytosolic (Fig. 4e, h, i).

**Fig. 6** γH2AX coprecipitates with PP1α and Cavin3 following UV stress. **a** A431 cells were treated with tautomycin (0.5 μM) for 3 h prior to UV irradiation. After a 4 h chase, cells extracts were western blotted for γH2AX and H2AX levels. β-Actin was blotted as the loading control. **b** γH2AX expression levels were quantified from three independent experiments where the data is presented as mean ± SD. Unt (−TTM): 100.0 ± 0.0%; Unt (+TTM): 155.1 ± 10.7%; UV (−TTM): 108.8 ± 31.5%; UV (+TTM): 215.2 ± 30.1%) where $*p = 0.0466$ was calculated using an ordinary one-way ANOVA. **c, d** GFP Trap assays of A431 cells transfected with GFP-PP1α or GFP (**c**), GFP-Cavin3 or GFP (**d**) followed by western bloting with an anti-γH2AX antibodies. Transfection efficiency was confirmed with GFP antibodies. α-Tubulin was used as the loading control. **e, g** PLA of the interaction of endogenous PP1α (**e**) or Cavin3 (**g**) with γH2AX upon UV radiation (shown as red dots). PLA signals alone were inverted to gray scale. Scale bar, 10 μm. Mean ± SD from three independent experiments are presented on the images. **f, h** The number and ratio of the PLA signals for PP1α (**f**) or Cavin3 (**h**) with γH2AX in the cytoplasm (gray) and nucleus (purple) were calculated and presented in the histograms and tables as mean ± SD from three independent experiments. **i** Phosphatase activity of PP1α in MCF-7 cells with or without mScarlet-PP1α (mS- PP1α) or/and pCB6-Cavin3 transfection. Relative phosphatase activity of control (%) (Ctrl: 100.0 ± 0.0%; mS-PP1: 123.4 ± 8.3%; Cavin3: 77.6 ± 10.0%; mS-PP1+Cavin3: 93.5 ± 7.4%) is presented as the mean ± SD from three independent experiments (Ctrl vs mS-PP1: $*p = 0.0209$, Ctrl vs Cavin3: $*p = 0.0260$, mS-PP1 vs mS-PP1+Cavin3: $**p = 0.0053$) using an ordinary one-way ANOVA

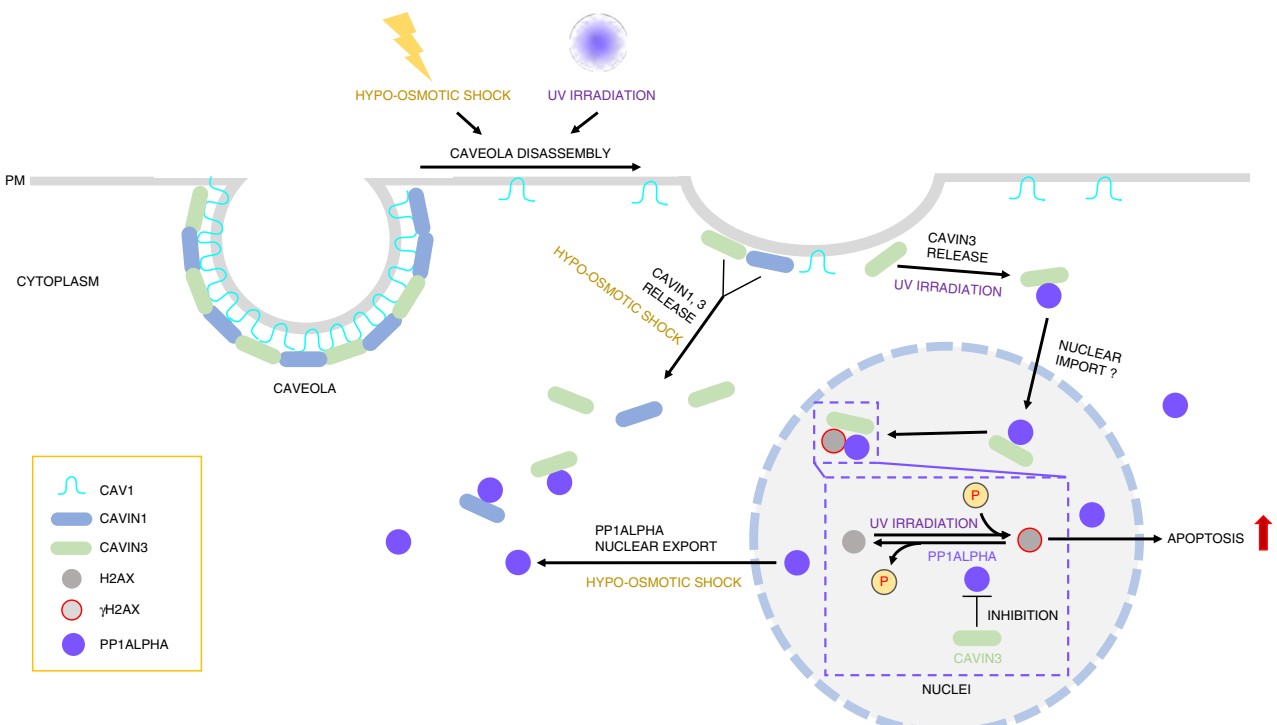

**Fig. 7** Schematic of the interaction and function of cavins and PP1α in response to stressors. In A431 cells (Cavin2-deficient), hypo-osmotic treatment induces redistribution of cavins and PP1α to the cytosol from plasma membrane caveolae and the nucleus, respectively, to allow interaction of these proteins. In addition, UV treatment led to the dissociation of Cavin3 from caveolae. We propose that the released Cavin3 may interact with the cytosol pool of PP1α first and be subsequently recruited to the nucleus as Cavin3-PP1α complex to associate with γH2AX. In the nucleus, Cavin3 exerts a proapoptotic role by promoting H2AX phosphorylation and its downstream apoptotic signaling through the interaction with and inhibition of PP1α, which has been identified as a direct phosphatase for γH2AX (phospho-H2AX) in this study

Furthermore, our studies reveal that interaction of cavins with PP1α can occur in response to different stimuli (hypo-osmotic versus UV treatment) to allow a cavin protein to interact in a quantitatively distinct manner with the same intracellular proteins. PP1 catalytic subunits typically combine with proteins that modulate their activity to direct them to distinct substrates[50–52]. More than 50 PP1-interacting proteins (PIPs) have been described. We now identify Cavin1 and Cavin3 as PP1α-interacting proteins. However, amino acid sequence analysis of the HR2 domain of Cavin1 and Cavin3 that was demonstrated to bind directly with PP1α, did not reveal an obvious PP1-binding consensus motif or SILK or MYPhoNE motif that plays an essential role in regulating PP1 activity[53]. Therefore, the modality of the interaction between the cavin proteins and PP1α remains to be determined.

Numerous reports have demonstrated involvement of PP1α in apoptosis[54–57] through interaction with Bad and Bcl-2[54]. Conversely, Cavin3 has also been shown to promote apoptosis[13] by increasing the apoptotic response of tumor cells to various stressors[31,32]. Our results suggest that Cavin3 and PP1α play opposite roles in the DNA damage response that is correlated with their protein expression in a number of cancers. Cavin3 is absence or downregulated due to promoter methylation[58–62], whereas PP1α is upregulated in hepatocellular[63] and oral squamous carcinomas[64]. Collectively, our findings suggest that the interaction between proapoptotic Cavin3 protein, and anti-apoptotic PP1α protein, is a death signal where further studies of this complex may provide insights into how cells evade apoptosis, an essential hallmark of cancer[65].

Multiple phosphatases have been implicated in negatively regulating γH2AX with emerging data suggesting redundancy as well as context-dependent specificity. Phosphorylation of H2AX is a very early event in the DNA damage response required for the assembly of DNA repair proteins at sites of damaged chromatin. Double strand breaks (DSBs) are also formed in the course of

DNA fragmentation in apoptotic cells[66] specifically in the final stages of this process. In this context, Cavin3-PP1α-γH2AX complex may act as a scaffold in facilitating chromatin remodeling and damage responsive factors at the sites of DSBs in terminally apoptotic cells to regulate the removal of γH2AX from this site through dephosphorylation by PP1α. We hypothesize that non-caveolar Cavin3 may bind to PP1α to regulate its phosphatase activity and consequently adjust the balance between cell death and cell survival following DNA damage. Further work is required to characterize the specific physiological substrates and regulation of the complex formed between Cavin3-PP1α-γH2AX.

The cavin interaction network generated by this work will lead to further understanding of caveolar function. The identification of a vast array of non-caveolar cavin-interacting proteins has provided evidence for involvement of these proteins in diverse cellular processes, such as metabolism and stress-signaling, which could not have been envisaged even a few years ago. These results will now allow us to begin to understand the role of caveolae in numerous diseases including cancer and the role of caveolae in protecting cells against stress.

## Methods

**Reagents**. Dulbecco's modified Eagle's medium (DMEM), glutamine, trypsin-EDTA were from Sigma-Aldrich (Castle Hill, New South Wales, Australia). Penicillin/streptomycin was from Invitrogen (Carlsbad, CA, USA). Fetal bovine serum (FBS) was from Life Technologies (Melbourne, Victoria, Australia). Protease Inhibitor Cocktail Set III was from Merck Millipore (Darmstadt, Germany) and PhoSTOP Phosphatase Inhibitor Cocktail was from Roche Diagnostics Australia (Castle Hill, NSW, Australia). Tautomycin was from Enzo Life Science (ALX-380-041-CO25, Michigan, USA).

**Antibodies**. The following antibodies were used: rabbit anti-CAV1 (dilution WB 1:3000) (610060, BD Biosciences, Franklin Lakes, New Jersey, USA), rabbit anti-Cavin1 antibody were raised as described previously by ref. [3] and was used for immunofluorescence (dilution IF/PLA 1:200), rabbit anti-Cavin3 (dilution WB 1:1000, IF/PLA 1:200) (16250-1-AP, ProteinTech Group, Chicago, IL, USA), rabbit anti-Cavin1 (WB 1:2000) (AV36965, Sigma-Aldrich, mouse anti-PRKCDBP/Cavin3 (dilution IF/PLA 1:200) (HOO112464-MO4, Novus Biologicals, LLC, Littleton, CO, USA) (dilution IF 1:200), rabbit anti-PP1α-FL (dilution WB 1:1000, IF/PLA 1:100) (sc-443 Santa Cruz, Biotechnologies, Inc, Dallas, TX, USA), mouse anti-PP1α (dilution WB 1:1000, IF/PLA 1:100) (G-4 71762, Santa Cruz, Biotechnologies), rabbit anti-cleaved caspase-3 (Asp175) antibody (dilution WB 1:1000) (9661, Cell Signaling Technology, Danvers, MA, USA), rabbit anti-caspase-3 (dilution WB 1:1000) (8G10, Cell Signaling Technology, Apoptosis and DNA damage H2AX (S139) and cleaved PARP and anti-GAPDH western blot cocktail antibody (dilution WB 1:1000) (ab131385, Abcam Australia, Pty, Ltd) rabbit anti-γH2AX (P139) (dilution WB 1:1000, IF/PLA 1:200) (ab2893, Abcam Australia, Pty, Ltd), rabbit anti-γH2AX antibody (dilution WB 1:1000) (ab11175, Abcam Australia), mouse anti-GFP (dilution WB 1:3000, IF/PLA 1:500) (11814460001, Roche), mouse anti-Tubulin (dilution WB 1:3000) (clone DM1A, T9026, Sigma-Aldrich), mouse anti-Cherry (dilution WB 1:2000) (5411-100, BioVision), and mouse anti-GAPDH (WB 1:2000) (AM4300, Thermo Fisher Scientific).

**DNA constructs**. Murine Cavin-1 constructs were described previously[67]. Murine Cavin-2 (Fantom 2 clone 9530015P22) and Cavin-3 (amplified from NIH-3T3 fibroblast cDNA) were cloned in either pEGFP vectors or mCherry vectors as previously described[3]. PP1alpha-GFP construct was kindly provided by the Trinkle-Mulcahy Laboratory, Canada.

**Plasmid preparation and cell-free expression**. Proteins were tagged with enhanced GFP (GFP), mCherry and cMyc (myc) tags, and cloned into cell-free-expression Gateway destination vectors: N-terminal GFP tagged (pCellFree_G03), N-terminal Cherry-cMyc (pCellFree_G07), and C-terminal Cherry-cMyc tagged (pCellFree_G08). Human RBPJ (BC020780) and MEF2C (BC026341) open reading frames (ORFs) were sourced from the Human ORFeome collection, version 1.1 and 5.1, and the Human Orfeome collaboration OCAA collection (Open Biosystems), as previously described[16] and cloned at the ARVEC facility, UQ Diamantina Institute. Translation competent *Leishmania tarentolae* extract (LTE) were prepared as previously described[16–18,68]. Protein pairs were co-expressed by adding 30 nM of GFP template plasmid and 60 nM of Cherry template plasmid to LTE and incubating for 3 h at 27 °C.

**Cell culture**. MCF-7 cells, a human adenocarcinoma cell line (ATCC HTB-22), were subjected to STR profiling (QIMR Berhofer Medical Research Institute), A431 cells (ATCC CRL-1555), a human skin/epidermis cell line and MDA-MB231 cells (ATCC HTB-26), were cultured in DMEM supplemented with 10% (vol/vol) FBS, 100 units/ml penicillin and 100 μg/ml streptomycin. All cell lines were subjected to routine mycoplasma testing. It should be noted that there is some discrepancy in the literature as to whether MCF-7 cells express CAV1 with some studies showing clear CAV1 mRNA and protein expression[69,70] whereas other studies show no CAV1 mRNA or protein expression[71,72]. Therefore, MCF-7 cells should be tested on a per laboratory basis for expression of CAV1 and the cavin proteins, respectively.

**Immunofluorescence**. In brief, MCF-7, MDA-MB231, or A431 cells seeded onto glass coverslips at 70% confluence were washed once in PBS and then fixed in 4% (vol/vol) PFA in PBS for 20 min at RT. Coverslips were washed three times in excess PBS and were permeabilized in 0.1% (vol/vol) Triton X-100 in PBS for 7 min and blocked in 1% (vol/vol) BSA (Sigma-Aldrich) in PBS for 30 min at RT. The primary antibodies were diluted in 1% (vol/vol) BSA in PBS and incubated for 1 h RT. Secondary antibodies (Molecular Probes) were diluted in 1% (vol/vol) BSA in PBS and incubated for 1 h RT. Washes were performed in PBS. Coverslips were rinsed in distilled water and mounted in Mowiol (Mowiol 488, Hoechst AG) in 0.2 M Tris-HCL, pH 8.5. The images were taken on a laser-scanning microscope (LSM 510 META, Carl Zeiss, Inc) using a ×63 oil lens, NA 1.4. Adjustments of brightness and contrast were applied using Image J software (NIH).

**SDS-PAGE and western blot analysis**. For SDS-PAGE, cells were harvested, rinsed in PBS and were lysed in lysis buffer containing 50 mM Tris pH 7.5, 150 mM NaCl, 5 mM EDTA pH 8.0, 1% Triton X-100 with protease and phosphatase inhibitors. Lysates were collected by scraping and cleared by centrifugation at 4 °C. The protein content of all extracts was determined using a BCA protein assay kit (Thermo Fisher Scientific, Victoria, Australia) using bovine serum albumin (BSA) as the standard. Thirty micrograms of cellular protein were resolved by 10% SDS-PAGE and were transferred to PVDF membranes (Millipore). Bound IgG was visualized with horseradish peroxidase-conjugated secondary antibodies and the Super Signal West Dura ECL detection reagent (Life Technologies) and was imaged using the ChemiDoc Imaging System (Bio–rad, Gladesville, New South Wales, Australia). Uncropped and unprocessed scans of the all blots presented in the main figures are available as a Supplementary Fig. 11 in the Supplementary Information.

**Proximity ligation assay (PLA)**. Detection of an interaction between proteins was assessed using the Duolink II Detection Kit (Sigma-Aldrich) according to the manufacturer specifications. The signal was visualized as a distinct fluorescent spot and was captured on an Olympus BX-51 upright Fluorescence Microscope using a ×60/1.35 oil lens. The number of PLA signals in a cell was quantified in Image J using a Maximum Entropy Threshold and Particle Analysis where 50 cells in each treatment group were analyzed for at least three independent experiments. RGB images were converted to black/white images with the Invert LUT from Image J.

**Stress experiments**. A431 or MDA-MB231 cells were plated on coverslips at 70% confluency. Cells were either left untreated or were treated with 70% hypo-osmotic media (70% H₂O in DMEM) for 10 min, or UV treatment for 2 min without media with a UV germicidal light source and allowed to recover in complete cell culture medium. All cells were fixed and processed using the Proximity Ligation assay as described.

**RNA interference**. Human Cavin3 Stealth siRNAs (set of 3-HSS174185, 150811, 150809), Human PP1α Stealth siRNAs (set of 3-HSS101089, 186096, 186097), and Stealth RNAi Negative Control Kit were purchased from Life Technologies Australia Pty Ltd. RNA oligonucleotides to Cavin3 or PP1α were transfected into cells at 24 h and 48 h after plating using Lipofectamine 3000 reagent (Invitrogen) with a ratio of 6 μl Lipofectamine to 150 pmol siRNA. Cells were split and harvested after 72–96 h for further analysis.

**Apoptosis assay**. Equal numbers of sub-confluent of control or Cavin3 or PP1α knockdown A431 cells were plated in 12-well dishes. Twenty-four hours later, cells were subject to UV C exposure for 2 min without media. Complete medium lacking phenol red was added to the cells that were left for an additional 4 h at 37 °C to recover. LDH release assay was measured in triplicate samples from 50 μL of conditioned media from each 12 well of cells using the Cytotoxicity Detection Kit (LDH) from Roche Diagnostics according to the manufacturer's instructions. Post nuclear supernatant from UV exposure cells were also prepared and were subjected to western blot analysis with antibodies to GFP (Roche), cleaved PARP (ABCAM), cleaved-caspase-3 (Cell Signaling), caspase-3 (Cell Signaling), Apoptosis and DNA damage H2AX (S139) and cleaved PARP and anti-GAPDH western blot cocktail antibody (Abcam), γH2AX (P139) (Abcam), mouse anti-GFP (Roche, Basel, Switzerland) and Tubulin (Abcam).

**Single-molecule spectroscopy**. Single-molecule spectroscopy was performed based on Leishmania cell-free lysates were prepared according to[16–18,68]. Where indicated MCF-7 cells were transiently cotransfected with GFP-PP1α and mCherry alone as the control, Cherry-Cavin1 or Cherry-Cavin3 constructs. A PNS fraction from the MCF-7 cells was prepared in PBS with protease and phosphatase inhibitors. Single-molecule coincidence measurements were performed using pairs of tagged proteins to ascertain their interaction. One protein of the pair was tagged with GFP, and the other with mCherry, and both were diluted to single-molecule concentrations (~1 nM). Two lasers, with wavelengths of 488 nm and 561 nm (to excite GFP and mCherry, respectively), were focused to a confocal volume using a 40×/1.2 NA water immersion objective. The fluorescence signal from the fluorophores was collected and separated into two channels with a 565 nm dichroic. The resulting GFP and mCherry signals were measured after passing through a 525/20 nm band pass and 580 nm long pass filter, respectively. The signal from both channels was recorded simultaneously with a time resolution of 1 ms, and the threshold for positive events was set at 50 photons/ms. The coincidence ratio (C) for each event was calculated as $C = mCherry/(GFP+mCherry)$, after subtracting a 6% leakage of the GFP signal into the mCherry channel. Coincident events corresponded to ~$0.25 < C < 0.75$. After normalizing for the total number of events (>1000 in all cases), a histogram of the C values for the protein pair was fitted with three Gaussians, corresponding to signals from solely GFP (green), coincidence (yellow), and solely mCherry (red).

**GFP Trap**. For each immunoprecipitation reaction, 3 × 10 cm dishes of MCF-7 or A431 cells expressing a GFP-tagged protein were extensively washed in ice-cold PBS three times followed by addition of RIPA buffer containing protease and phosphatase inhibitors (Roche). Tubes were placed on ice for 20 min with extensively pipetting every 10 min cell lysates were centrifuged at 20,000 × g for 10 min at 4 °C. Lysate-supernatants were transferred to a pre-cooled tube and the pellet was discarded. GFP Trap beads were equilibrated in 0.5 ml of ice-cold RIPA buffer and were spun down at 2.500 × g for 2 min at 4 °C. Beads were washed two more times with 500 µl RIPA buffer. In total, 1–2 mg of lysate-supernatant was added to equilibrated GFP Trap beads and were incubated for 1 h, 4 °C with constant mixing. Tubes were spun at 2.500 × g for 2 min at 4 °C. GFP Trap beads were washed three times with 500 µl ice-cold RIPA buffer. One hundred microliters of 2× SDS sample buffer was then added to the GFP Trap beads and were boiled for 10 min at 95 °C. The beads were collected by centrifugation at 2.500 × g for 2 min and SDS-PAGE was performed with the supernatant.

**Biotin ligase transfection and purification**. MCF-7 cells at 80% confluency were transfected with BirA-IRES-GFP (control) or BirA-IRES-Cavin3 using Lipofectamine 3000 according to the manufacturer's instructions. After 3 h, the cell culture medium was replaced with 2 ml of fresh medium containing 10% FBS serum and 50 µM biotin. Cells were further incubated for 18 h at 37 °C. Cells were washed three times with ice-cold PBS and were lysed in RIPA buffer (50 mM Tris-HCl pH 7.5, 150 mM NaCl, 0.1% SDS, 1% Triton X-100 and 5 mM EDTA) containing phosphatase and protease inhibitors (Roche). Cell extracts were lysed with a 25-gauge needle and syringe, 10–20 times and were incubated on ice for 20 min. Samples were spun at 14,000 × g for 10 min at 4 °C and the supernatant was removed. High affinity strepavidin agarose beads (Thermo Scientific) with washed three times with RIPA buffer and was added to 2 mg of total cell extract that was left to rotate at 4 °C overnight. The agarose beads were pelleted at 2500 × g and washed three times in RIPA buffer. Eighty microliters of 2× SDS-PAGE sample buffer containing DTT was added to the beads and boiled 5 min at 95 °C. All samples were pelleted at 2500 × g for 5 min at RT before loading on an SDS-PAGE gel. Western blot membranes were blocked with 5% BSA in TBST for at least 1 h and were extensively wash (four times) with TBST 10 min prior to detection with Clarity Western ECL substrates (Bio-Rad Laboratories, Hercules, California, USA).

**ALPHAScreen**. ALPHAScreen was performed as previously described[16–18], using the cMyc detection kit and Proxiplate-384 Plus plates (PerkinElmer). The plates were incubated for 45 min at room temperature, followed by the addition of streptavidin-coated donor beads and incubation in the dark for 45 min at room temperature. The ALPHAScreen signal was measured on an Enivision Plate Reader (PerkinElmer) using manufacturer's recommended settings. Data from three independent experiments was analyzed in GraphPad Prism version 6.0. The luminescence intensities plotted are averages over three independent expressions/experiments where for each experiment, the signal intensity from a negative control, GFP alone in solution was substracted. An average of all the normalized data for both configuration of each protein pair (GFP-protein A/protein B-Cherry or GFP protein B/protein A-Cherry) was calculated to provide the binding index where values above 2000 were considered background. From previous experiments[16–18], a threshold of above 2000 (background) selected positive interactions and was therefore employed in these studies.

**AlphaLISA assay**. GFP-tagged HR1, HR2 and HR1+HR2, Cavin3 and Cavin1, and Cherry-tagged PP1α were co-expressed in an LTE system[73]. Following their expression, the protein mixture was diluted 100 times with buffer A (25 mM HEPES, 50 mM NaCl, 0.1% BSA and 0.01% Nonidet P-40). The AlphaLISA assay

was carried out in an Optiplate-384 Plus plate, using anti-GFP AlphaLISA acceptor and streptavidin donor beads. Alpha beads were prepared according to the protocol provided by PerkinElmer. The alpha acceptor and donor beads stocks were diluted to 100 µg/mL in 1X AlphaLISA Universal assay buffer. The biotinylated mCherry nanobody (100 nM) was added into microplate wells followed by addition of 15 µL protein mixture (PP1α-Cherry and GFP-Cavin3) and (PP1α-Cherry and pair-FKBP) as a negative control. All samples were prepared in triplicate. The acceptor beads (5 µL) were added to each well and the mixture was incubated for 1 h at room temperature. Finally, 5 µL of donor beads were added to the samples under subdued light, mixed gently and incubated 30 min at RT. The AlphaLISA signal was detected with a Tecan microplate reader using the following setting: Filter: AlphaLISA, Excitation time: 180 ms, Integration time: 300 ms.

**Mass spectrometry of GFP Trap pulldowns**. Pulldowns were eluted in SDS-PAGE sample buffer and separated on 10% SDS-PAGE gels to 2 mm only. Protein visualization, excision of bands and in-gel trypsin digest were performed using a semi-automated method as described in ref. [74]. Peptides were analyzed using either Agilent nano-LC QTOF with Spectrum Mill for database searching as previous described[75], or using the following method. Thermo Scientific Q Exactive Plus Orbitrap Mass Spectrometer coupled with Easy-nLC 1000 and EASY-spray ion source was used to analyze the digested peptides. Samples were loaded onto an EASY-Spray PepMap RSLC C18 2 µm column (25 cm × 75 µm ID), with a nano-viper acclaim C18 guard (75 m × 2 cm). A 62 min method was run using a combination of Buffer A (0.1% Formic acid) and Buffer B (0.1% Formic acid: Acetonitrile). A two-step gradient was run comprising a 10 min gradient from 3 to 10% Buffer B and a 30 min gradient from 10 to 32% Buffer B. Flow rate was at 3 L/min. The mass spectrometer was programmed to acquire a full MS resolution of 70,000 with an ACG target of 3e6 with a maximum injection time of 100 ms. The MS scan range was from 350 to 1400 m/z. MS/MS was set to acquire a resolution of 35,000 with an ACG target of 5e5 and maximum injection time of 110 ms. The loop count was set to 20 with a dynamic exclusion after 10 s.

Raw data were processed with Proteome Discoverer (Thermo, 2.0.0.802). Selected modifications included fixed carbamidomethylation of cysteine and variable oxidized methionine. Results were searched against the Human SwissProt database (v2015-09-16) using the Sequest HT node. Trypsin was selected for enzyme digest, with three maxima missed cleavages allowed. The precursor mass tolerance was set at +/−10 ppm and fragment mass tolerance was 0.08 Da. False discovery rate was set to 0.05 using the percolator node. Search parameters were defined as a rapid search using trypsin digestion enzyme, iodoacetamide cysteine alkylation and all entries in the database. Proteins were considered identified if there were two or above peptides identified with a 99% confidence and a 1% global false discovery rate (FDR). Network representation of selected biological processes and pathways was performed using Ingenuity Pathway analysis (QIAGEN Bioinformatics) content version 18841524. The mass spectrometry proteomics data have been deposited to the ProteomeXchange Consortium via the PRIDE partner repository with the dataset identifier PXD014081.

**Nano HPLC, mass spectrometry, and protein identification for BioID**. For BioID/MS, the human cell protein, trypsin digested, extracts were analyzed by nanHPLC/MS MS/MS on a Shimadzu Prominance Nano HPLC (Japan) coupled to a Triple Tof 5600 mass spectrometer (ABSCIEX, Canada) equipped with a nano electrospray ion source. Sixteen microliters of each extract was injected onto a 50 mm × 300 µm C18trap column (Agilent Technologies, Australia) at 30 µl/min. The samples were de-salted on the trap column for 5 min using 0.1% formic acid (aq) at 30 µL/min. The trap column was then placed in-line with the analytical nano HPLC column, a 150 mm × 100 µm 300SBC18, 3.5 µm (Agilent Technologies, Australia) for mass spectrometry analysis. Linear gradients of 1–40% solvent B over 50 min at 300 nL/min flow rate, followed by a steeper gradient from 40 to 80% solvent B in 5 min were used for peptide elution. Solvent B was held at 80% for 6 min for washing the column and returned to 1% solvent B for equilibration prior to the next sample injection. Solvent A consisted of 0.1% formic acid (aq) and solvent B contained 90/10 acetonitrile/0.1% formic acid (aq). The ionspray voltage was set to 2400 V, declustering potential (DP) 100 V, curtain gas flow 25, nebulizer gas 1 (GS1) 12 and interface heater at 150 °C. The mass spectrometer acquired 500 ms full scan TOF-MS data followed by 20 by 50 ms full scan product ion data in an Information Dependant Acquisition, IDA, mode. Full scan TOFMS data were acquired over the mass range 300–1400 and for product ion ms/ms 80–1400. Ions observed in the TOF-MS scan exceeding a threshold of 120 counts and a charge state of +2 to +5 were set to trigger the acquisition of product ion, ms/ms spectra of the resultant 20 most intense ions. The data were acquired and processed using Analyst TF 1.6.1 software (ABSCIEX, Canada). Proteins were identified by database searching using ProteinPilot v4.5 (ABSCIEX, Canada) against the UniProt_Sprot_20130205 Human Protein database (~40,532 entries searched, FDR of 1%). The mass spectrometry proteomics data have been deposited to the ProteomeXchange Consortium via the PRIDE partner repository with the dataset identifier PXD014094.

**Electron microscopy**. For electron microscopy, A431 cells were either left untreated or were treated with UV for 2 min as described and then further incubated for 30 min before fixation for epon embedding or were treated with 70%

hypo-osmotic medium (70% water in DMEM, 10 min). Sections were cut perpendicular to the culture substratum. Processing and quantitation of the density of caveolae was performed on over 30 images for each condition captured at a primary magnification of 25kx on a Jeol 1010 or Jeol 1011. Three sets of images from three independent experiments were analyzed for the presence of caveolae on the apical surface of the cell and normalized to the length of surface sampled (over 150 µm for each set of images). Imaging and quantification for all EM studies were performed in a blinded fashion.

**Phosphatase activity assay.** MCF-7 cells transfected with mScarlet-PP1α or/and pCB6-Cavin3 DNA were lysed in lysis buffer (20 mM Tris-HCl, pH 7.4, 132 mM NaCl, 10% Glycerol, 1% Triton X-100, and a protease inhibitor cocktail tablet from Roche). Lysates containing 500 µg protein (made up to 500 µL by lysis buffer) were precleared with protein A-coupled sepharose beads (20 µL). After centrifuging at $2500 \times g$ for 1 min, supernatants were collected and immunoprecipitated with anti-PP1α antibody (ab137512, Abcam) overnight at 4 °C. Protein A (20 µL) was added for another 3 h incubation. The beads were washed three times with lysis buffer and then resuspended in 50 µL of reaction buffer (50 mM Tris-HCl at pH 7.0 containing 0.1 mM CaCl₂, 125 µg/mL BSA and 0.05% Tween 20) provided by the RediPlate 96 EnzChek serine/threonine phosphatase activity assay kit (R33700, Thermo Fisher). Reaction buffer containing immunoprecipitates was then added into the wells incorporated with 50 µM phosphatase substrate 6,8-difluoro-4-methylumbelliferyl phosphate for a 30 min incubation at 37 °C in the dark. Fluorescence was then measured at an excitation wavelength of 355 nm and an emission wavelength of 460 nm using a TECAN microplate reader. Relative phosphatase activity was calculated as a percentage of the control.

**Gene expression analysis in MCF-7 cells.** Human Cavin 1, 2 and 3, and Caveolin-1 gene expression was analyzed using the Taqman Gene Expression assay (Life Technologies Australia, Applied Biosystems Division). RT PCR primers used were Hs00396859_m1 Cavin1, Hs00190538_m1 Cavin2, Hs04194683_s1 Cavin3, Hs00971716_m1 Cav1, and Hs00427620_m1 TBP as the endogenous control. Total RNA was extracted from cells using the TRI reagent (Sigma-Aldrich) according to the manufacturer's protocol. RNA was further purified using a mini-Uneasy kit (QIA-GEN) according to manufacturer's instructions and quantified using a NanoDrop ND-1000 spectrophotometer. cDNA was synthesized from 1 µg of total RNA for cell culture and using Superscript III primed by oligo dT (Geneworks), according to the manufacturer's instructions (Invitrogen). Target cDNA levels were compared with qRT-PCR in 25 µl reactions containing Taqman PCR master mix (Roche Molecular Systems) 1× Assay-on-Demand Taqman primers and the equivalent of 0.3 µL cDNA. Using an ABI Prism 7500 (Applied Biosystems), PCR was conducted over 45 cycles of 95 °C for 15 s and 60 °C for 1 min, preceded by an initial 95 °C for 10 min. Expression levels were normalized to HPRT1 as determined from the ratio of delta CT values. All results are expressed as mean ± SEM from five independent replicates. Statistical analyses were performed using GraphPad Prism Version6a (California, USA). All qRT-PCR data were analyzed using a one-way ANOVA with a Tukey's multiple correction test.

**Statistical analyses.** Statistical analyses were conducted using Microsoft Excel and Prism (GraphPad). Error bars represent either standard deviation (SD) or standard error of the mean (SEM) for at least three independent experiments, as indicated in the figure legends. Statistical significance was determined either by two-tailed Student's $t$-test or by one-way ANOVA, as indicated in the figure legends. Significance was calculated where $*p < 0.05$, $**p < 0.01$, $***p < 0.001$, and $****p < 0.0001$.

**Reporting summary.** Further information on research design is available in the Nature Research Reporting Summary linked to this article.

## Data availability
The source data underlying Supplementary Figures 2 and 4, and Supplementary Data 1 (mass spectrometry data from BioID and GFP-Trap experiments) are provided as a source data file titled "NCOMMS-18-13078A_Source_Data" in a single excel file. All reagents and further experimental data are available from the corresponding author upon reasonable request. Proteomics data for BioID and GFP-Trap experiments that supports the findings of this study have been deposited to the ProteomeXchange Consortium via the PRIDE repository separately with the dataset identifier PXD014094 (BioID) and PXD014081 (GFP-Trap).

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

## Acknowledgements

We would like to thank Nicholas Ariotti for valuable discussion. Thanks to Sachini Fonseka and Ye-Wheen Lim for proofreading the article. This work was supported by fellowships and grants from the National Health and Medical Research Council of Australia (to R.G.P., grant numbers 569542 and 1045092) and to (R.G.P., A.S.Y., and K.A., grant number 1037320). M.M.H. was supported by a Fellowship from the Australian Research Council (FT120100251). Confocal microscopy was performed at the Australian Cancer Research Foundation (ACRF)/Institute for Molecular Bioscience (IMB) Dynamic Imaging Facility for Cancer Biology, established with funding from the ACRF. The authors acknowledge the facilities, and the scientific and technical assistance, of the Australian Microscopy & Microanalysis Research Facility at the Centre for Microscopy and Microanalysis, The University of Queensland and the Proteomic Facility at UQ Diamantina Institute and Translational Research Institute. We acknowledge Alun Jones and the IMB Mass Spectromewtry Facility. The BirA/BioID constructs were kindly provided by the Burke Laboratory, Singapore and the PP1alpha construct was kindly provided by Laura Trinkle-Mulcahy, Canada.

## Author contributions

K.A.M., Y.W., and R.G.P. conceived the study. K.A.M. performed the biotin/mass spectrometry experiments, interpreted the data, and wrote the paper with R.G.P., A.Y., and K.A. Y.W. performed all PP1/Cavin1-3/γ-H2AX experiments and wrote that section of the paper, Y.G. and E.S. performed single-molecule analysis and the ALPHAScreen, V. T. made the PP1 phosphorylation constructs, S.O. made the BirA-Cavin1 and Cavin3 constructs, T.H. and N.M. made the PP1α cell-free-expression vector, J.R. and M.M.H. performed mass spectrometry analysis, C.F. performed Electron Microscopy, and S.V.M. performed the AlphaLISA analysis. All authors provided useful insights and commented on the paper.

## Additional information

**Competing interests:** The authors declare no competing interests.

