## [Peer Review File · Nature Communications]

Reviewers' comments:

Reviewer #1 (Remarks to the Author):

Review of NCOMMS-18-13078 entitled "Identification of intracellular Cavin target proteins reveals a role for Cavin-PP1alpha interactions in regulation of apoptosis" by Parton and colleagues.

Summary: The authors use several methods to investigate protein-protein interactions for Cavin proteins. They performed BioID-based proximity labeling of Cavin3 in vivo (MCF-7 cell line), with the potential advantage of permitting the detection of weak and/or transient interactors without requiring interactions survive post-lysis, along with affinity purification/APMS(GFP-trap), and an ALPHAScreen chemiluminescent proximity assay to establish binary interactions in vitro cell-free expression. From these 3 techniques, they generated a list of potential intracellular Cavin-interacting proteins. The authors then explore the interaction they identify between Cavin and PP1a, which is confirmed by coIP of tagged proteins and Proximity ligation assay (PLA) in MCF-7 cells. They investigate the functional relevance of the Cavin-PP1a interaction, and show how the interaction responds to caveola disassembly, with A431 cells induced with hypoosmotic media showing an increase association of Cavin with PP1a. To further investigate this link, Cavin3 relocalization from caveolae to the cytosol and association with PP1a is examined in A431 cells treated with UV. siRNA knockdown in A431 cells suggests Cavins may play opposite roles to PP1a in DNA damage and apoptosis responses. GFP-Cavin3 and GFP-PP1a IPs show UV induces interaction with the DNA damage response histone γ H2AX.

Specific comments

Overall, the complementary interaction data is interesting and convincing, and the cell biology is pretty solid. They combine lots of different approaches and the results are consistent. Don't see anything in dire need of clarification or additional experiments and I think it would be fine to publish the cell data as is.

If I am really nit-picky, the one experiment were I would like to see more primary data is Figure 1C. They don't show the signal for PP1a itself and I would be curious to see if there is a complete overlap with the PLA signal aka if all the cytosolic PP1a interacts with any of the cavins or if there may be differences between the three family members.

For figure 4, the authors could show or comment on previously published data, if available, if PP1a also redistributes into the cytosol in hypo-osmotic conditions in cells that lack caveolae, e.g. the MCF7 cells they use for their first set of experiments. I am wondering if cavins are required for the translocation and/or stabilization of PP1a in the cytosol.

Given the proposed role of cavin3 in DNA damage and cell survival, is there a difference in UV sensitivity between MCF7 and A431 cells and if so, is expression of cavin3 in MCF7 enough to revert that effect. This would be another way to show that the effect of cavin3 is independent of caveolae as MCF7 don't express caveolin.

Tables 1 and Figure 1 present data from BioID and ALPHAScreen. Much of this could instead be presented in supplemental figures. The authors may benefit from combining BioID, GFP-trap, and ALPHAScreen data into one figure, as well as indicating which ORFs were chosen from literature alone.

Figure 2a&b should display western blots for both the input and GFP-IP.

Reviewer #2 (Remarks to the Author):

Caveolae formation at the plasma membrane requires the coat protein caveolin-1 as well as a family of adaptors called cavins. The cavins were initially identified as proteins with alternate roles in the cell including as transcription factors. Caveolae flatten and release cavins upon mechanical stress and it remains unknown what the functional role of cavins is upon release due to mechanical stretch. This paper identifying non-caveolar interactions of cavins and potential roles in apoptosis is therefore interesting and topical. However, the switch in models from hypotonic shock to UV has a number of inconsistencies and the overall story is not convincing. There is no mechanistic data included for the hypotonic shock side of the story and no clear mechanistic link to the UV-induced stress story. Further, the authors do not convincingly show that the cavin released from caveolae is affecting PP1 nor is it really clear how cavin is affecting PP1 activity. Is it sequestering PP1a out of the nucleus or is the cavin3 somehow inactivating nuclear PP1a. Finally, the data supporting cavin3-PP1a interaction as apoptotic regulators is not convincing nor is it clear how this is actually working. These concerns are outlined in more detail below and seriously dampen enthusiasm for this manuscript.

1. The proximity ligation assay is very intriguing but it would be important to validate the Cav1-dependence of this interaction by co-expressing Cav1 in the MCF7 cells in the PLA assay and single molecule coincidence assay (Fig 2) to show that Cav1 sequestration of the cavins impacts its ability to interact with PP1a.

2. Hypotonic shock in A431 cells in Figure 4 shows the redistribution of cavin1 from PM to cytosol and PP1 from the nucleus to cytosol. Both associate with cytosolic aggregates that do not appear to colocalize even though the PLA signal does increase. What are these aggregates that are quite distinct from the diffuse cytosolic distribution of cavin-GFPs and why is there minimal colocalization? Finally, the increased PLA signal and these experiments require some critical controls (Cav1, cavin siRNA) to show the Cav1/cavin specificity of the redistribution of PP1 from the nucleus.

3. Figures 5 and 6 switch to UV-induced membrane damage and it is not clear why similar experiments exploring the role of Cavin-PP1 interaction were not studied in models of mechanical stretch-induced apoptosis. Again the loss of caveolae and increased cavin-PP1 interaction is a correlation. And if cavin3 is pro-apoptotic through interaction with and inhibition of PP1a then why does Cavin3 siRNA still reduce apoptotic H2AXp and cleaved caspase-3 in the absence of PP1a? This argues that the pro-apoptotic role of cavin-3 is PP1a independent.

4. The cavin3-PP1a PLA interaction in response to hypoosmotic shock is clearly cytoplasmic but in response to UV the Cavin3-PP1 PLA signal appears to be nuclear? This has to be clarified - and validated by alternative approaches - and raises serious questions as to why the authors are combining the hypoosmotic shock and UV data. I understood the argument to be that cavin released from caveolae sequesters PP1 out of the nucleus (see Fig 4AB) and thereby prevents dephosphorylation of H2AX. Based on the pulldown data in Fig 6 A, it would appear that in response to UV, cavin-3 translocates to the nucleus where it regulates the phosphatase activity of PP1a and thereby phosphorylation of H2AX.

Minor

1. Additional information should be provided on how the 47 proteins chosen for alpha screen analysis were selected.
2. The supplemental table has a duplicate entry for PKM and should be checked for others.
3. Total H2AX blots should be included.

Reviewer #3 (Remarks to the Author):

This manuscript provides proteomic data of cavin-3 interacting partners with validation focused on one

of these interactors, PP1a. Proteomics uses a number of approaches, including a BIOID/MS approach with a BirA-tagged cavin-3 and an ALPHAscreen approach. Validation experiments examined cavin-3 interaction with PP1a in response to hypotonic stress and UV irradiation, both of which were shown to increase interaction of cavin-3 with PP1a.

There are a number of intriguing findings in this paper. The interaction of cavin-3 with PP1a is novel. There is also a number of other potential interactors that are of high interest (e.g., p53). On the downside, there are also a great many loose ends that greatly detract from the paper and some follow up experiments that are needed to better validate the results (see specific concerns below).

In general, the statistical analysis was appropriate for the data presented.

Concerns regarding the proteomics:

The BIOID/MS with the Cavin-3 BirA and the Cavin-3 BirA + Cavin-1 did not produce a common hit. This lack includes PPP1CA, which was identified in Cavin-3 alone, but not in the Cavin-1+3 proteinID. What does this mean? The hits from Cavin-3 + Cavin-1 also did not appear to have been tested in the ALPHAscreen. Were they tested? Why include the Cavin-3+1 data in Table 1 if the data is ignored in the rest of the manuscript? Is the thought that cavin-1 displaces cavin-3 only binders? If so, how does cavin-3 associate with PPP1CA in A431 cells, which express endogenous cavin-1?

Most of the top hits in the Cavin-3 BIOID/MS were also not reported in the ALPHAscreen results (proteins such as HSPA5, ALDOA, SQSTM1, UGDH, PP1A, PKM, HNRNPK and PPP1CC all had as many or more peptides than PPP1CA in the BIOID). No rationale for their exclusion was given.

For the ALPHAscreen a readout of 2000 was used as the demarcation between positive and negative results. How was this value arrived at? PPP1CA was barely above this line (2321 in the Cavin-3 experiment) and was much less than top hits such as p53 (17864 in the Cavin-3 experiment).

Concerns regarding validation:

I am not clear on why cavin-2 was excluded from the analysis halfway through the results. The authors state that they excluded cavin-2 from further analysis because the mCherry cavin-2 pulldown with GFP PP1a was weaker than that with mCherry cavin-1 and mCherry cavin-3; however, the pulldowns with endogenous PP1a showed equivalently strong interaction with cavin-1 and cavin-2, but only weak interaction with cavin-3. Furthermore, the PLA results with cavin-2 were as strong as those with cavin-1 and stronger than those with cavin-3.

Fig 3 nicely identifies that the domain (HR2) of cavin-3 mediates an interaction with PP1a, but there is no equivalent analysis for cavin-1 or cavin-2. Both cavin-1 and cavin-2 have a similar domain structure with an HR1 and HR2 domain and it should be tested whether PP1a binding is a common feature of the HR2 domains.

Why were A431 cells chosen as the cell line with endogenous expression of caveolin and cavins for validation? While I understand the convenience in Fig 1 of using a cell line like MCF-7 cells, which have no endogenous caveolin or cavin expression, the choice of a transformed cancer line and A431 in particular for validation is not at all clear. Why not use a normal human mammary epithelia cell line to complement MCF-7 (which is a breast cancer line)? Caveolin and cavins have been reported to be expressed in normal mammary epithelia. Use of A431 cells is also worrying in that they are an epidermoid cancer cell line, which expresses very high levels of EGFR. EGFR activation has been implicated in alterations of caveolae dynamics. Furthermore, in vivo, epidermal cells are regularly exposed to UV irradiation and undergo dehydration as part of the keratinization process. The degree

to which A431 cells maintain any unique responses to UV irradiation and osmotic stress (Fig 4-5) is not clear to me and is not discussed.

The methods state that a confocal microscope was used to acquire fluorescent images; however, data for the PLA assays do not appear to be confocal slices and it is not clear whether the foci of PLA fluorescence are in the cytosol or the nucleus. Foci need to be assessed in cytosol and the nucleus separately because gH2AX operates in the nucleus and if cavins alter PP1a activity toward H2AX, this presumably occurs in the nucleus, not the cytosol.

Figure 4 shows redistribution of both cavin-1 and cavin-3 from the plasma membrane to the cytosol and PP1a from the nucleus to the cytosol in response to hypotonic treatment. There is a PLA signal following treatment that is higher than that before treatment; however, while there appears to be areas of higher density of PP1a and cavins in the cytosol, these areas do not overlap (no colocalization). Given the much lower PLA signals in the A431 experiments (4-fold over background) as compared to the PLA signals in the MCF7 system (100-fold), a functional output of PP1 should be added. PP1 binds to at least 200 proteins. Many of these interactions inhibit the activity of PP1 and cavin-3 may do likewise. There are commercial assay kits for PP1 activity.

For the EM studies of Fig 5A, representative images should be provided in either Fig 5 or in the supplement. The Y-axis of Fig 5A should be in units of caveolae per mm of membrane. Imaging and quantification should be done in a blinded fashion (not stated in methods). Total mm of membrane quantified should be given for all trials.

Given the EM data in Fig 5, EM data comparing caveolae/mm needs to be shown for the hypotonic experiments of Fig 4

Figure 5 needs to show confocal images of cells with and without UV treatment. Hypotonic treatment caused redistribution of PP1 from the nucleus to the cytosol (Fig 4). What happens with UV treatment? This is particularly important because Fig 6 shows a PLA reaction of cavin-3 with gH2AX that appears to be in the nucleus (though the PLA images, as with other PLA images, do not appear to be a confocal slice and what appears to be nuclear cavin-3/gH2AX complexes may be in the cytosol above/below the nucleus).

A direct test of whether PP1a +/- cavin-3 dephosphorylates gH2AX is needed.

Discussion concerns:

3/4 of PP1a interacting proteins use a RVxF motif to associate with PP1a. Many interacting proteins complement the RVxF motif with a SILK motif or a MyPhoNE motif, both of which also participate in PP1a binding. Does cavin-1, -2, and/or -3 have any of these motifs? If present, are these motifs in the HR2 domains of these proteins?

General manuscript concerns: This paper has the appearance having been rushed to submission and has many figure and writing errors.

Many reference numbers wrong throughout much of the paper (e.g., p 3, 3rd paragraph)

In the results for the PLA experiment on p6, the first paragraph states "all subsequent experiments focused on the interaction of PP1a with cavin-1 and cavin-3"; however, the next paragraph shows PLA data with cavin-2. If PLA data is shown for cavin-2, why is there no coincidence detection data for cavin-2 alongside that for cavin-1 and cavin-3?

In Fig 6A, why is there no GFP in the input?

sFig 1A is missing the MDA MB 231 lane (it is mentioned in the figure legend)

sFig 3 is missing the panel for UV treated cavin1/PP1a cells

The figure legend for sFig 4 states three color images (cavin3, PP1a, DAPI), but the DAPI image is missing from the figure.

Reviewer #4 (Remarks to the Author):

This ms. by McMahon et al investigates the role of Cavin proteins in cell signalling, based on hypothesis that release of Cavins from caveolae, during cell stress, liberates signalling functions. Using a comprehensive suite of approaches, BiotinID, GFP Trap, PLA, Alpha Screen – the authors develop an extensive data set for Cavin binding proteins. Importantly, this is largely done in MCF-7 cells to enforce Cavin expression into the cytosol, mimicking a signalling environment. Focusing on one specific interaction Cavin1/3 with PP1 alpha – they confirm this, map binding domains and investigate a functional relationship for this interaction in the regulation of DNA-damage response and apoptosis.

Overall, this study provides an important resource for Cavin-binding proteins, with emphasis on their roles in signalling. Here, the data support the authors' conclusions. My main (only) criticism of the work centers upon the regulation of DNA-damage by the PP1alpha/Cavin-3 interaction (detailed below).

The authors make conclusions that the PP1 alpha/Cavin 3 complex is regulating UV induced DNA-damage (γH2AX) and apoptosis via opposite of RNAi of either protein on the extent of DNA-damage/apoptosis (more when PP1alpha is knocked down, less when Cavin-3 is knocked down). These differential effects may be due to the PP1alpha/Cavin3- complex, however it remains entirely plausible that knockdown of either protein is having these effects through independent means. Given that the authors have defined the HR2 domain (in Cavin 3) as being responsible for the PP1alpha interaction, they should carry out rescue expts. (in the DNA-damage/apoptosis expts) to determine with re-introduction of the WT Cavin (but not HR2 mutant) reverts the effects of Cavin-3 knockdown – if so, this would significantly strengthen their model.

It is important to measure the extent (% of cells dying by) of apoptosis, as opposed to using the LDH readout shown in 5H – the way it is presented can be prone to overestimation, e.g an incremental increase in cell death from 1 to 2% would, if everything is linear, provide a 200% increase in LDH levels, if the 1% cell death is set at 100% LDH. Measuring actual % cell death is a much more accurate way to determine any effect of PP1alpha, Cavin-3 knockdown

Point-by-point response to reviewers

Reviewer #1 (Remarks to the Author):

Specific comments:

Overall, the complementary interaction data is interesting and convincing, and the cell biology is pretty solid. They combine lots of different approaches and the results are consistent. Don't see anything in dire need of clarification or additional experiments and I think it would be fine to publish the cell data as is.

We appreciate the positive comments of the reviewer.

If I am really nit-picky, the one experiment where I would like to see more primary data is Figure 1C. They don't show the signal for PP1a itself and I would be curious to see if there is a complete overlap with the PLA signal aka if all the cytosolic PP1a interacts with any of the cavins or if there may be differences between the three family members.

This is an excellent question and we have now included additional experiments that contribute to a more extensive characterization of the distribution of the partner proteins, particularly PP1 α , in the revised manuscript (see revised Figure 4A-B and Figure S10A). Little colocalization was observed for endogenous cavins and PP1 α under control conditions consistent with their different localization under steady state conditions (Figure 4A-B, upper panels; Figure S10, upper panel). In contrast, under stress conditions, cavins were redistributed from caveolae to the cytosol and nucleus, where they exhibited colocalization (Figure 4A-B, lower panels; Figure S10, lower panel) and increased interaction with PP1 α (Figure 4D-G; Figure 5A-B). Unfortunately, direct comparison of endogenous proteins using PLA and IF was impossible because both of the methods rely on antibody-based detection of the proteins of interest.

For figure 4, the authors could show or comment on previously published data, if available, if PP1a also redistributes into the cytosol in hypo-osmotic conditions in cells that lack caveolae, e.g. the MCF7 cells they use for their first set of experiments. I am wondering if cavins are required for the translocation and/or stabilization of PP1a in the cytosol.

We know of no published data that has looked at the localization of PP1 α under hypotonic condition in cells specifically lacking caveolae, but here we have now shown that PP1 α is redistributed into the cytosol in cavin-deficient MCF7 cells following hypo-osmotic treatment (revised Figure S7). This strongly suggests that hypo-osmotic stress-induced translocation of PP1 α is a cavin/caveolae-independent event.

Given the proposed role of cavin3 in DNA damage and cell survival, is there a difference in UV sensitivity between MCF7 and A431 cells and if so, is expression of cavin3 in MCF7 enough to revert that effect. This would be another way to show that the effect of cavin3 is independent of caveolae as MCF7 don't express caveolin.

We have now compared the sensitivity of MCF-7 cells +/- Cavin3-GFP versus A431 cells to UV using the LDH assay. However, as presented below (Figure R1), we observed that A431 cells were more resistant to UV irradiation than MCF-7 cells. This could be related to many different mechanisms, independent of the caveolar system. Therefore, to further address this question, we have performed additional experiments in MCF-7 cells now presented in the revised Figure 4H. MCF-7 cells were transfected with GFP-cavins alone or with additional mCherry-CAV1 and Flag-Cavin1 (two

proteins absolutely required for caveolar formation). Caveola formation and cavin sequestration through the co-expression of CAV1 and Cavin1 in these cells resulted in decreased interaction of PP1 α with GFP-cavins, strongly suggesting that the effect of Cavin3 in UV-induced apoptosis is independent of caveolae and consistent with our proposed model: Cavin3 must be released from caveolae.

Figure R1. LDH release in A431 cells (caveolae proficient) and MCF-7 cells (caveolae deficient) with Cavin3 re-expression. A431 cells and MCF-7 cells with or without Cavin3 re-expression were subjected to UV irradiation for 2 min. After a 4 h chase, LDH assays were performed to assess the plasma membrane integrity.

Tables 1 and Figure 1 present data from BioID and ALPHAScreen. Much of this could instead be presented in supplemental figures. The authors may benefit from combining BioID, GFP-trap, and ALPHAScreen data into one figure, as well as indicating which ORFs were chosen from literature alone.

We have now combined Table 1 and Figure 1 together as suggested into Table S1. The ORFs that were identified from the literature are indicated in the combined table (orange column) and an additional Table S2 that is a compiled list of potential interacting proteins of Cavin1 and Cavin3 with a list of the experimental evidence databases where these interactions were described together with published references, that were used to form the basis for candidate selection for the ALPHAScreen.

Figure 2a&b should display western blots for both the input and GFP-IP.

We now present these data in a revised Figure 2A and B.

Reviewer #2 (Remarks to the Author):

Caveolae formation at the plasma membrane requires the coat protein caveolin-1 as well as a family of adaptors called cavins. The cavins were initially identified as proteins with alternate roles in the cell including as transcription factors. Caveolae flatten and release cavins upon mechanical stress and it remains unknown what the functional role of cavins is upon release due to mechanical stretch. This paper identifying non-caveolar interactions of cavins and potential roles in apoptosis is therefore interesting and topical. However, the switch in models from hypotonic shock to UV has a number of inconsistencies and the overall story is not convincing. There is no mechanistic data included for the hypotonic shock side of the story and no clear mechanistic link to the UV-induced stress story. Further, the authors do not convincingly show that the cavin released from caveolae is affecting PP1 nor is it really clear how cavin is affecting PP1 activity. Is it sequestering PP1a out of the nucleus or is the cavin3

somehow inactivating nuclear PP1a. Finally, the data supporting cavin3-PP1a interaction as apoptotic regulators is not convincing nor is it clear how this is actually working. These concerns are outlined in more detail below and seriously dampen enthusiasm for this manuscript.

1. The proximity ligation assay is very intriguing but it would be important to validate the Cav1-dependence of this interaction by co-expressing Cav1 in the MCF7 cells in the PLA assay and single molecule coincidence assay (Fig 2) to show that Cav1 sequestration of the cavins impacts its ability to interact with PP1a.

We have now performed the requested experiment as shown (revised Figure 4H). As seen in representative confocal microscopy images, the expression of Cav1 in MCF7 cells recruits GFP-cavins (see enlarged images showing the colocalization of mCherry-CAV1 and GFP-tagged Cavin1 or Cavin3), and parallel PLA results shows a significant decrease in the interaction of cavins with PP1 α in these cells. This provides additional support for our model that release of cavins from caveolae is required for the interaction of cavin proteins with PP1 α and that depending on the cell stressor, this interaction can occur in the cytosol and nucleus of these cells.

2. Hypotonic shock in A431 cells in Figure 4 shows the redistribution of cavin1 from PM to cytosol and PP1 from the nucleus to cytosol. Both associate with cytosolic aggregates that do not appear to colocalize even though the PLA signal does increase. What are these aggregates that are quite distinct from the diffuse cytosolic distribution of cavin-GFPs and why is there minimal colocalization? Finally, the increased PLA signal and these experiments require some critical controls (Cav1, cavin siRNA) to show the Cav1/cavin specificity of the redistribution of PP1 from the nucleus.

Consistently, our experiments demonstrate that Cavin1 and Cavin3 as well as PP1 α are redistributed to the cytosol following hypotonic shock in A431 cells. As requested by the reviewer, we have now presented new confocal microscopy images showing the subcellular localisation of endogenous Cavin1, Cavin3 and PP1 α by immunofluorescence in the revised Figure 4A-B, providing additional support for our model. In response to hypotonic treatment, the redistribution of cavin proteins and PP1 α to the cytosol from the plasma membrane and the nucleus respectively, was observed. In addition, the colocalization between cavins and PP1 α in the cells with hypotonic treatment was confirmed by confocal microscopy (Figure 4A-B, lower panel). We do not believe that these are artefactual aggregates as we are detecting the endogenous proteins, but the exact nature of these complexes is still unknown. Importantly, we have now carried out a set of control experiments in which we have depleted/knocked down expression of Cavin3 and PP1 α using siRNA as presented in Figure S8. As shown in the revised Figure S8, we see a clear loss of PLA signal after knockdown of either protein validating the PLA approach.

3. Figures 5 and 6 switch to UV-induced membrane damage and it is not clear why similar experiments exploring the role of Cavin-PP1 interaction were not studied in models of mechanical stretch-induced apoptosis. Again the loss of caveolae and increased cavin-PP1 interaction is a correlation. And if cavin3 is pro-apoptotic through interaction with and inhibition of PP1a then why does Cavin3 siRNA still reduce apoptotic H2AXp and cleaved caspase-3 in the absence of PP1a? This argues that the pro-apoptotic role of cavin-3 is PP1a independent.

Hypo-osmotic shock as a published method (Sinha, *et al*, 2011; reference no. 9; Gambin *et al.*, 2014; reference no. 8) was applied as a mechanical stress to study cavin release and subsequent interaction with PP1 α . We now provide compelling electron microscopy quantitation studies of caveolae numbers following hypo-osmotic treatment in A431 cells that shows a significant decrease in caveolar number as demonstrated in previously published studies (Sinha *et al.*, 2011; Gambin *et al.*, 2014).

In addition, we chose UV irradiation as a “non-mechanical” stress for this study. We focused on the functional consequences of the interaction between Cavin3 and PP1 α following UV irradiation because of the previous link between Cavin3 and cellular response to UV (as stated in the manuscript, page 9, line 15, Lee *et al.*, 2008; reference no. 35 and Lee *et al.*, 2011, reference no. 36) and importantly, the significance of UV as the cause and regulator during the development of skin cancer (Chan *et al.*, 2003, reference no. 32; Liao *et al.*, 2015; reference no. 33, Lin *et al.*, 2014, reference no. 34) (stated in revised manuscript, page 8, line 11). We further extended our mechanistic studies to the function of the Cavin3-PP1 α complex in UV-induced apoptosis in A431, a skin cancer cell line, to expand our understanding of a possible role for this complex in UV induced signalling events related to skin cancer.

We believe that Cavin3 exerts a pro-apoptotic role in cells in a PP1 α -dependent manner for the following reasons: First, densitometry of western blot results in our study showed that consistently, in the absence of PP1 α , depletion of Cavin3 (cells with double PP1 α -siRNA and Cavin3-siRNA knockdowns) caused no significant changes in the expression levels of γ H2AX (Figure 5E, panel a), and apoptotic markers including cleaved caspase3 (Figure 5G, panel b) and cleaved PARP (Figure 5G, panel a), compared to control cells (scramble) following UV irradiation, which strongly argues that the apoptotic role of Cavin3 is indeed PP1 α dependent. In contrast, Cavin3 depletion alone in A431 cells specifically and significantly reduced γ H2AX (Figure 5E, panel a), cleaved caspase3 (Figure 5G, panel b) and cleaved PARP (Figure 5G, panel a) levels following UV irradiation, indicating that PP1 α is required for Cavin3 to regulate UV-induced apoptosis. In addition, our overexpression studies demonstrate that Cavin3 overexpression can neutralize the anti-apoptotic effect induced by PP1 α overexpression (revised Figure S10C-F). Collectively, these data suggest that PP1 α is an important intracellular target of Cavin3 in the promotion of apoptosis following UV irradiation and that the proapoptotic role of Cavin3 is PP1 α dependent.

4. The cavin3-PP1a PLA interaction in response to hypoosmotic shock is clearly cytoplasmic but in response to UV the Cavin3-PP1 PLA signal appears to be nuclear? This has to be clarified - and validated by alternative approaches - and raises serious questions as to why the authors are combining the hypoosmotic shock and UV data. I understood the argument to be that cavin released from caveolae sequesters PP1 out of the nucleus (see Fig 4AB) and thereby prevents dephosphorylation of H2AX. Based on the pulldown data in Fig 6A, it would appear that in response to UV, cavin-3 translocates to the nucleus where it regulates the phosphatase activity of PP1a and thereby phosphorylation of H2AX.

We thank the reviewer for this comment. We have used two independent stimuli (we did not combine these stimuli) to test the hypothesis that release of cavin proteins from caveolae is crucial for the proposed interaction. Both stimuli (hypo-osmotic shock as a mechanical stimuli and UV irradiation as a “non-mechanical stimuli”) result in an increase in the interaction between the cavin proteins and PP1 α . Furthermore, these two stimuli do result in a different localization of the complex, as the reviewer points out. Nevertheless, we believe that the approaches used in this study are complementary and point towards a general role for caveola disassembly in facilitating this interaction. We further provide evidence in the revised Figure 6A and B using a specific PP1 α inhibitor tautomycin for 3 hours prior to UV irradiation, that PP1 α in response to UV is most likely the phosphatase responsible for inhibiting H2AX phosphorylation levels in these cells. In addition, the revised Figure 6G demonstrates relative phosphatase activity of PP1 α measured in MCF-7 cells with or without Cavin3 co-expressed, that clearly demonstrates that Cavin3 significantly inhibit the phosphatase activity of PP1 α . Collectively, these new data demonstrate that released Cavin3 is a new inhibitory regulator for PP1 α that consequently controls the balance between cell death and

cell survival following UV-induced DNA damage through the regulation of H2AX phosphorylation in these cells.

Minor

1. Additional information should be provided on how the 47 proteins chosen for alpha screen analysis were selected.

This is now included in the text of the manuscript p5. In addition, Table S2 is now included that is a compiled list of published potential interacting proteins for Cavin1 and Cavin3 that was used to select candidate proteins for testing by the ALPHAScreen including experimental evidence for the interactions and published references.

2. The supplemental table has a duplicate entry for PKM and should be checked for others. This has been removed with the combination of the tables in Figure 1 with the Supplementary Table.

We apologise for the error and thank the reviewer for picking this up. This error has now been corrected and we have rechecked the table with a new combined Table S1.

3. Total H2AX blots should be included.

Total H2AX blots have now been included in the revised Figure 5D and Figure S10 that show equal levels to total H2AX in all samples blotted.

Reviewer #3 (Remarks to the Author):

This manuscript provides proteomic data of cavin-3 interacting partners with validation focused on one of these interactors, PP1a. Proteomics uses a number of approaches, including a BIOID/MS approach with a BirA-tagged cavin-3 and an ALPHAScreen approach. Validation experiments examined cavin-3 interaction with PP1a in response to hypotonic stress and UV irradiation, both of which were shown to increase interaction of cavin-3 with PP1a.

There are a number of intriguing findings in this paper. The interaction of cavin-3 with PP1a is novel. There is also a number of other potential interactors that are of high interest (e.g., p53). On the downside, there are also a great many loose ends that greatly detract from the paper and some follow up experiments that are needed to better validate the results (see specific concerns below). In general, the statistical analysis was appropriate for the data presented.

Concerns regarding the proteomics:

The BIOID/MS with the Cavin-3 BirA and the Cavin-3 BirA + Cavin-1 did not produce a common hit. This lack includes PPP1CA, which was identified in Cavin-3 alone, but not in the Cavin-1+3 proteinID. What does this mean? The hits from Cavin-3 + Cavin-1 also did not appear to have been tested in the ALPHAScreen. Were they tested? Why include the Cavin-3+1 data in Table 1 if the data is ignored in the rest of the manuscript? Is the thought that cavin-1 displaces cavin-3 only binders? If so, how does cavin-3 associate with PPP1CA in A431 cells, which express endogenous cavin-1?

We agree with the comments of the reviewer and we have now removed the Cavin1+3 data from the table as it detracted from the general focus of the manuscript. We have now included a revised

Table S1 that includes all BioID/MS and Cavin3-GFP/MS in A431 and MCF-7 cells, ALPHAScreen data as well as candidate proteins that were chosen from literature searches.

Most of the top hits in the Cavin-3 BIOID/MS were also not reported in the ALPHAScreen results (proteins such as HSPA5, ALDOA, SQSTM1, UGDH, PP1A, PKM, HNRNPK and PPP1CC all had as many or more peptides than PPP1CA in the BIOID). No rationale for their exclusion was given.

We have now added more explanation in the manuscript text (page 5, first paragraph in ALPHAScreen/*In vitro* expression analysis result section) and have further included Table S2 which is a compiled list of published potential interacting proteins for Cavin1 and Cavin3 including experimental evidence to support the interactions as well as the paper references that was used to select candidate proteins for testing by the ALPHAScreen.

For the ALPHAScreen a readout of 2000 was used as the demarcation between positive and negative results. How was this value arrived at? PPP1CA was barely above this line (2321 in the Cavin-3 experiment) and was much less than top hits such as p53 (17864 in the Cavin-3 experiment).

We apologise for the omission and have now expanded the description of the methods and quantitation. As now stated on page 6, top paragraph, the luminescence intensities plotted are averages over three independent expressions/experiments where for each experiment, the signal intensity from a negative control, GFP alone in solution, was subtracted. An average of all the normalized data for both configuration of each protein pair (GFP-protein A/protein B-Cherry OR GFP-protein B/protein A-Cherry) was calculated to provide the binding index. From previous studies (Sierecki *et al.*, 2014; reference no. 19) and (Moustaqil *et al.*, 2018; reference no. 20), a threshold of above 2000 selected positive interactions and was therefore employed also in these studies. In the case of Cavin3, this threshold corresponds to the top 20-25% of the binding intensities. In addition, ALPHAScreen intensity response as a function of Cavin3 protein dilution was further measured for Cavin3 binding proteins as well as for each cavin protein with PP1 α which is now presented in revised Figure S3.

Concerns regarding validation:

I am not clear on why cavin-2 was excluded from the analysis halfway through the results. The authors state that they excluded cavin-2 from further analysis because the mCherry cavin-2 pulldown with GFP PP1a was weaker than that with mCherry cavin-1 and mCherry cavin-3; however, the pulldowns with endogenous PP1a showed equivalently strong interaction with cavin-1 and cavin-2, but only weak interaction with cavin-3. Furthermore, the PLA results with cavin-2 were as strong as those with cavin-1 and stronger than those with cavin-3.

Cavin2 data has now been excluded from Figure 2 as there was consistently a much weaker and less stable interaction with PP1 α as demonstrated by GFP Trap pulldown experiments as well as ALPHAScreen. We also discovered, through the course of these experiments, that Cavin2 is not a universal component of caveolae in cultured cells where we have observed no expression of Cavin2 in cell lines such as A431 cells used in these studies as well as others such as HeLa cells. We now state this on page 7, first paragraph. We therefore focused our studies on Cavin1 and Cavin3.

Fig 3 nicely identifies that the domain (HR2) of cavin-3 mediates an interaction with PP1a, but there is no equivalent analysis for cavin-1 or cavin-2. Both cavin-1 and cavin-2 have a similar domain structure with an HR1 and HR2 domain and it should be tested whether PP1a binding is a common feature of the HR2 domains.

We thank the reviewer for this insightful suggestion. Additional experiments have now been performed using an ALPHALisa assay presented in the revised Figure 3 to test whether PP1 α binding is a common feature of the HR2 domain of Cavin1. We have now tested direct interaction between *in vitro* synthesised Cavin1 domain mutants and PP1 α as presented in revised Figure 3, and have demonstrated that constructs containing the HR2 domain of Cavin1 when coexpressed with PP1 α , also show highest associated with PP1 α , suggesting that PP1 α binding is a common feature and mediated by the HR2 domain of Cavin1 and Cavin3. Furthermore, this is the first report of a protein that binds specifically to the HR2 domains of Cavin1 and Cavin3.

Why were A431 cells chosen as the cell line with endogenous expression of caveolin and cavin for validation? While I understand the convenience in Fig 1 of using a cell line like MCF-7 cells, which have no endogenous caveolin or cavin expression, the choice of a transformed cancer line and A431 in particular for validation is not at all clear. Why not use a normal human mammary epithelia cell line to complement MCF-7 (which is a breast cancer line)? Caveolin and cavin have been reported to be expressed in normal mammary epithelia. Use of A431 cells is also worrying in that they are an epidermoid cancer cell line, which expresses very high levels of EGFR. EGFR activation has been implicated in alterations of caveolae dynamics. Furthermore, *in vivo*, epidermal cells are regularly exposed to UV irradiation and undergo dehydration as part of the keratinization process. The degree to which A431 cells maintain any unique responses to UV irradiation and osmotic stress (Fig 4-5) is not clear to me and is not discussed.

A431 cell line has been widely used for the study of caveolae due to their abundance of caveolae at the plasma membrane (as discussed on page 8, Ahmed *et al.*, 2010, reference no 29, Lee *et al.*, 2009, reference no 30). In addition, as an epidermoid cancer cell line, A431 has been chosen as a model for the mechanistic studies of cell apoptosis following UV irradiation (Thomsen *et al.*, 2002; reference no. 31, Chan *et al.*, 2003; reference no 32). Therefore, we believe that the A431 cell line is an ideal cell model for us to explore caveolae/cavin dynamics and function under “non-mechanical” UV stress. In addition, we performed PLA in an additional cell line, MDA-MB 231, and observed the upregulation of the interaction between Cavin3 and PP1 α in both cytosol and nucleus following UV irradiation, suggesting that UV-induced interaction between Cavin3 and PP1 α is a general effect in different cell lines (revised Figure S9).

The methods state that a confocal microscope was used to acquire fluorescent images; however, data for the PLA assays do not appear to be confocal slices and it is not clear whether the foci of PLA fluorescence are in the cytosol or the nucleus. Foci need to be assessed in cytosol and the nucleus separately because γ H2AX operates in the nucleus and if cavin alters PP1a activity toward H2AX, this presumably occurs in the nucleus, not the cytosol.

We now show in the revised Figure 6E and F with representative confocal microscopy images, that PLA signals for γ H2AX and Cavin3 or γ H2AX and PP1 α upon UV treatment occurs predominantly in the nucleus as indicated by the overlap between DAPI to mark the nuclei in these cells.

Figure 4 shows redistribution of both cavin-1 and cavin-3 from the plasma membrane to the cytosol and PP1a from the nucleus to the cytosol in response to hypotonic treatment. There is a PLA signal following treatment that is higher than that before treatment; however, while there appears to be areas of higher density of PP1a and cavin in the cytosol, these areas do not overlap (no colocalization). Given the much lower PLA signals in the A431 experiments (4-fold over background) as compared to the PLA signals in the MCF7 system (100-fold), a functional output of PP1 should be added. PP1 binds to at least 200 proteins. Many of these interactions inhibit the activity of PP1 and cavin-3 may do likewise. There are commercial assay kits for PP1 activity.

This is an excellent suggestion which led us to carry out a new series of experiments to directly test the effect of Cavin3 on PP1 α . We now include the phosphatase assay results (revised Figure 6G) showing that Cavin3 inhibits the phosphatase activity of PP1 α . These results allow us to propose a new model for the regulation of PP1 α by UV released Cavin3 protein.

For the EM studies of Fig 5A, representative images should be provided in either Fig 5 or in the supplement. The Y-axis of Fig 5A should be in units of caveolae per mm of membrane. Imaging and quantification should be done in a blinded fashion (not stated in methods). Total mm of membrane quantified should be given for all trials.

We have changed the axis to units of caveolae per mm of membrane as requested. Indeed, imaging and quantification for all EM studies were performed in a blinded fashion which is now stated in the *Electron Microscopy Methods* section on p21.

Given the EM data in Fig 5, EM data comparing caveolae/mm needs to be shown for the hypotonic experiments of Fig 4.

We have now included EM studies of the caveolar number following hypo-osmotic treatment in A431 cells which is presented in revised Figure S6. We demonstrate a significant decrease in caveolar number following hypo-osmotic treatment in A431 cells as previously described (Sinha *et al.*, 2011; reference no. 8).

Figure 5 needs to show confocal images of cells with and without UV treatment. Hypotonic treatment caused redistribution of PP1 from the nucleus to the cytosol (Fig 4). What happens with UV treatment? This is particularly important because Fig 6 shows a PLA reaction of cavin-3 with γ H2AX that appears to be in the nucleus (though the PLA images, as with other PLA images, do not appear to be a confocal slice and what appears to be nuclear cavin-3/ γ H2AX complexes may be in the cytosol above/below the nucleus).

This is an important point and we have now shown by confocal microscopy that Cavin3 is redistributed into the nucleus with γ H2AX after UV treatment in the revised Figure 5A and Figure 6F.

A direct test of whether PP1 α +/- cavin-3 dephosphorylates γ H2AX is needed.

To further strengthen these studies, as presented in the revised Figure 6A and B, A431 cells were treated with 0.5 μ M tautomycin (a specific PP1 α inhibitor) for 3 h prior to UV irradiation. Cell extracts were then blotted for γ H2AX and total H2AX. In response to UV irradiation, a significant upregulation of γ H2AX levels was observed in tautomycin-treated cells compared to untreated cells, whereas total H2AX remained unchanged. This result demonstrates that PP1 α is most likely the phosphatase that dephosphorylates γ H2AX in these cells. Furthermore, relative phosphatase activity of PP1 α measured in MCF-7 cells with and without Cavin3 reexpression clearly demonstrated that Cavin3 can inhibit the phosphatase activity of PP1 α , suggesting that Cavin3 may function as an inhibitory regulator for PP1 α and thereby promote H2AX phosphorylation in these cells. This is now presented in the revised Figure 6G.

Discussion concerns:

3/4 of PP1a interacting proteins use a RVxF motif to associate with PP1a. Many interacting proteins complement the RVxF motif with a SILK motif or a MyPhoNE motif, both of which also participate in PP1a binding. Does cavin-1, -2, and/or -3 have any of these motifs? If present, are these motifs in the HR2 domains of these proteins?

After extensive searching, we have not been able to find a RVxF, SILK or MyPhoNE motif in the cavin proteins, with particular consideration of the HR2 domain of Cavin1 and Cavin3. The HR2 domain of Cavin1 and Cavin3 does have many surface exposed arginine residues that may participate in PP1 α binding. This is now stated in the Discussion on p13, middle paragraph.

General manuscript concerns: This paper has the appearance having been rushed to submission and has many figure and writing errors. Many reference numbers wrong throughout much of the paper (e.g., p 3, 3rd paragraph)

We apologise for these errors. We have endeavoured to correct all the mistakes and have the manuscript checked by additional readers.

In the results for the PLA experiment on p6, the first paragraph states "all subsequent experiments focused on the interaction of PP1a with cavin-1 and cavin-3"; however, the next paragraph shows PLA data with cavin-2. If PLA data is shown for cavin-2, why is there no coincidence detection data for cavin-2 alongside that for cavin-1 and cavin-3?

Cavin2 showed a much weaker or less stable interaction with PP1 α as shown in the GFP-Trap results (Figure 2A-B) and as stated, during the course of our studies, we found that cell model line A431 cells do not express Cavin2. We have therefore removed all GFP/Trap pulldown and PLA experiments related to Cavin2 and PP1 α and have focused the remainder of the paper on Cavin1 and Cavin3. (See our discussion about Cavin2 above and on page 7, first paragraph).

In Fig 6A, why is there no GFP in the input?

We have now provided images that include the GFP with longer exposure time in the input in the revised Figure 6C.

sFig 1A is missing the MDA MB 231 lane (it is mentioned in the figure legend)

This has now been corrected.

sFig 3 is missing the panel for UV treated cavin1/PP1a cells.

This has now been corrected.

The figure legend for sFig 4 states three color images (cavin3, PP1a, DAPI), but the DAPI image is missing from the figure.

This has now been corrected.

Reviewer #4 (Remarks to the Author):

This ms. by McMahon et al investigates the role of Cavin proteins in cell signalling, based on hypothesis that release of Cavins from caveolae, during cell stress, liberates signalling functions. Using a comprehensive suite of approaches, BiotinID, GFP Trap, PLA, Alpha Screen – the authors develop an extensive data set for Cavin binding proteins. Importantly, this is largely done in MCF-7 cells to enforce Cavin expression into the cytosol, mimicking a signalling environment. Focusing on one specific interaction Cavin1/3 with PP1 alpha – they confirm this, map binding domains and investigate a functional relationship for this interaction in the regulation of DNA-damage response and apoptosis.

Overall, this study provides an important resource for Cavin-binding proteins, with emphasis on their roles in signalling. Here, the data support the authors' conclusions. My main (only) criticism of the work centers upon the regulation of DNA-damage by the PP1alpha/Cavin-3 interaction (detailed below).

The authors make conclusions that the PP1 alpha/Cavin 3 complex is regulating UV induced DNA-damage (γ H2AX) and apoptosis via opposite of RNAi of either protein on the extent of DNA-damage/apoptosis (more when PP1alpha is knocked down, less when Cavin-3 is knocked down). These differential effects may be due to the PP1alpha/Cavin3- complex, however it remains entirely plausible that knockdown of either protein is having these effects through independent means. Given that the authors have defined the HR2 domain (in Cavin 3) as being responsible for the PP1alpha interaction, they should carry out rescue expts. (in the DNA-damage/apoptosis expts) to determine with re-introduction of the WT Cavin (but not HR2 mutant) reverts the effects of Cavin-3 knockdown – if so, this would significantly strengthen their model.

We thank the reviewer for this suggestion. We have now included additional evidence in the revised manuscript showing PP1 α as a phosphatase for γ H2AX (revised Figure 6A-B) and the inhibitory effect of Cavin3 on PP1 α phosphatase activity (revised Figure 6G). These data further support our hypothesis that Cavin3 promotes UV-induced apoptosis through the interaction with, and inhibition of PP1 α . However, it was technically difficult to compare the effects of HR1 and HR2 domains on apoptosis due to the difference in the stabilities of the HR1 and HR2 Cavin3 constructs in cells. HR1 truncate is mainly localised in the cytosol where it is continuously degraded (Tillu et al., 2015; reference no. 48), whereas HR2 is distributed in the nucleus and shows higher stability. Also, the mislocalization of HR1 and HR2 domains suggest that these truncated Cavin3 constructs might not reflect the real function of wild type Cavin3.

In addition, the aim of this study is to provide a comprehensive molecular insight of non-caveolar cavins in cellular pathways. Although PP1 α was selected for further functional investigation in this study, our ALPHAScreen data (Figure 1D) has revealed multiple cavin-interacting proteins that are related to apoptotic pathways, such as p53, suggesting cavin-related apoptosis effects might not be exclusively caused by PP1 α and may also involve interactions with these proteins that may or may not occur within the HR2 domains of the cavins.

It is important to measure the extent (% of cells dying by) of apoptosis, as opposed to using the LDH readout shown in 5H – the way it is presented can be prone to overestimation, e.g an incremental increase in cell death from 1 to 2% would, if everything is linear, provide a 200% increase in LDH levels,

if the 1% cell death is set at 100% LDH. Measuring actual % cell death is a much more accurate way to determine any effect of PP1alpha, Cavin-3 knockdown

We thank the reviewer for this suggestion. All LDH release results have been redone to include a measurement of the total LDH levels from each sample to allow the calculation of the LDH release (% of total LDH) and is now presented in revised Figure 5H (knockdown experiments) and Figure S10G (overexpression experiments).

REVIEWERS' COMMENTS:

Reviewer #1 (Remarks to the Author):

The authors have addressed the main criticisms, and the paper is now suitable for publication.

Reviewer #2 (Remarks to the Author):

The authors have done an admirable job of addressing the previous comments and the manuscript is better constructed, with a clearer focus on the role of cavin3 in the response to UV stress. The issue of cytoplasmic or nuclear localization of the Cavin3-PP1a-H2AX complex remains unresolved. The PLA interaction of both PP1a and Cavin3 with H2AX (Fig 6 EF) appear to be more nuclear localized than the cavin3-PP1a PLA shown in Fig 5A. Does this mean that Cavin3-PP1a complexes form in the cytoplasm and then translocate to the nucleus to interact with H2AX? Is this the basis for the differential localization of the Cavin3-PP1a complex in response to hypotonic vs UV stress? Instead of describing the localization as "cytoplasmic and nuclear" it would be important to better define the cytoplasmic vs nuclear localization of the complex as determined by PLA. Determining if indeed, in response to UV, H2AX recruits the Cavin3-PP1 complex to the nucleus would be important to better define the proposed mechanism.

Reviewer #3 (Remarks to the Author):

The resubmitted manuscript provides a proteomic interactome for cytosolic cavin proteins with validation for one protein, PP1. The proteomic data used a combination of a discovery approach in the form of a BirA BIOID/MS protocol, followed by confirmatory approaches using GFP-trap, ALPHAscreen, and proximity ligation (PLA). Components of the PP1 phosphatase complex were identified in each method. The authors then investigated the significance and context of this interaction, showing that the HR2 domain of cavin proteins mediate the interaction with PP1 and that this interaction inhibits PP1 phosphatase activity. Importantly, the authors also show that this interaction only occurs under stress conditions that displace cavin proteins from plasma membrane caveolae and that one function of the cavin-PP1 interaction is to suppress the ability of PP1 to dephosphorylate H2AX, thereby likely contributing to the pro-apoptotic activity of at least cavin-3. Overall, the manuscript provides important new information that improves our understanding of the interplay between caveolae and stress signaling. Cavin-3 in particular has been shown to promote apoptosis under stress conditions and the presented work provides one avenue by which this may work. With regards to the revision, the current submission has been greatly improved over the original and all of my major concerns have been adequately addressed. Statistical handling of data was appropriate.

Peter Michaely

Reviewer #4 (Remarks to the Author):

The authors have comprehensively addressed all points raised in my initial review.

REVIEWERS' COMMENTS:

Reviewer #1 (Remarks to the Author):

The authors have addressed the main criticisms, and the paper is now suitable for publication.

We appreciate the positive feedback of the reviewer.

Reviewer #2 (Remarks to the Author):

The authors have done an admirable job of addressing the previous comments and the manuscript is better constructed, with a clearer focus on the role of cavin3 in the response to UV stress. The issue of cytoplasmic or nuclear localization of the Cavin3-PP1a-H2AX complex remains unresolved. The PLA interaction of both PP1a and Cavin3 with H2AX (Fig 6 EF) appear to be more nuclear localized than the cavin3-PP1a PLA shown in Fig 5A. Does this mean that Cavin3-PP1a complexes form in the cytoplasm and then translocate to the nucleus to interact with H2AX? Is this the basis for the differential localization of the Cavin3-PP1a complex in response to hypotonic vs UV stress? Instead of describing the localization as "cytoplasmic and nuclear" it would be important to better define the cytoplasmic vs nuclear localization of the complex as determined by PLA. Determining if indeed, in response to UV, H2AX recruits the Cavin3-PP1 complex to the nucleus would be important to better define the proposed mechanism.

We appreciate the comments of the reviewer. This is an important question but to elucidate the precise sequence of events leading to the formation of the cytosolic Cavin3-PP1 α complex and the nuclear Cavin3-PP1 α - γ H2AX complex will require further experimentation outside the scope of this work. However, as the reviewer requested, we have performed additional quantification of the number and ratio of PLA signals in the nucleus and cytoplasm and presented the new results in the revised figures 4f-i, 5c, 6f and 6h. In addition, we have now stated the ratio (%) of nuclear or cytoplasmic localized PLA signals to describe the distribution of complexes in the revised manuscript (page 8, 9 and 11). Furthermore, we have now clarified the proposed model in the revised "Discussion" section in the manuscript (page 13) as below:

"Interestingly, Cavin3-PP1 α - γ H2AX complexes appear more concentrated in the nucleus (Figure 6e-f) than Cavin3-PP1 α complexes after UV treatment (Figure 5a). This suggests that the formation of Cavin3-PP1 α - γ H2AX complex could be a dynamic process, whereby the released Cavin3 may interact with the cytosolic pool of PP1 α first and subsequently translocate into the nucleus as a Cavin3-PP1 α complex to associate with γ H2AX (Figure 7). Future experimentation will be required to elucidate the precise sequence of events leading to the formation of the cytoplasmic Cavin3-PP1 α complex and the nuclear Cavin3-PP1 α - γ H2AX complex."

Regarding the role of γ H2AX, we have included additional discussion on page 13:

"However, we believe that the differential distribution of Cavin3, PP1 α or Cavin3-PP1 α complexes under hypo-osmotic treatment and UV stress is independent of γ H2AX as γ H2AX is also induced in the nucleus in response to hypo-osmotic treatment (49) yet Cavin3-PP1 α complex is mainly cytosolic (Figure 4e and 4h-i)."

Reviewer #3 (Remarks to the Author):

The resubmitted manuscript provides a proteomic interactome for cytosolic cavin proteins with validation for one protein, PP1. The proteomic data used a combination of a discovery approach in the form of a BirA BIOD/MS protocol, followed by confirmatory approaches using GFP-trap, ALPHAscreen, and proximity ligation (PLA). Components of the PP1 phosphatase complex were identified in each method. The authors then investigated the significance and context of this interaction, showing that the HR2 domain of cavin proteins mediate the interaction with PP1 and that this interaction inhibits PP1 phosphatase activity. Importantly, the authors also show that this interaction only occurs under stress conditions that displace cavin proteins from plasma membrane caveolae and that one function of the cavin-PP1 interaction is to suppress the ability of PP1 to dephosphorylate H2AX, thereby likely contributing to the pro-apoptotic activity of at least cavin-3. Overall, the manuscript provides important new information that improves our understanding of the interplay between caveolae and stress signaling. Cavin-3 in particular has been shown to promote apoptosis under stress conditions and the presented work provides one avenue by which this may work. With regards to the revision, the current submission has been greatly improved over the original and all of my major concerns have been adequately addressed. Statistical handling of data was appropriate.

Peter Michaely

We appreciate the positive feedback of Reviewer 3.

Reviewer #4 (Remarks to the Author):

The authors have comprehensively addressed all points raised in my initial review.

We appreciate the positive feedback of Reviewer 4.